# Discrete vulnerability to pharmacological CDK2 inhibition is governed by heterogeneity of the cancer cell cycle

Vishnu Kumarasamy[1], Jianxin Wang [1], Michelle Roti[1], Yin Wan[1], Adam P. Dommer[1], Hanna Rosenheck[1], Sivasankar Putta[2], Alec Trub[3], John Bisi[3], Jay Strum[3], Patrick Roberts[3], Seth M. Rubin [2], Costakis Frangou[1], Karen McLean [4], Agnieszka K. Witkiewicz[1] ✉ & Erik S. Knudsen [1] ✉

Cyclin dependent kinase 2 (CDK2) regulates cell cycle and is an emerging target for cancer therapy. There are relatively small numbers of tumor models that exhibit strong dependence on CDK2 and undergo G1 cell cycle arrest following CDK2 inhibition. The expression of P16INK4A and cyclin E1 determines this sensitivity to CDK2 inhibition. The co-expression of these genes occurs in breast cancer patients highlighting their clinical significance as predictive biomarkers for CDK2-targeted therapies. In cancer models that are genetically independent of CDK2, pharmacological inhibitors suppress cell proliferation by inducing 4N cell cycle arrest and increasing the expressions of phospho-CDK1 (Y15) and cyclin B1. CRISPR screens identify CDK2 loss as a mediator of resistance to a CDK2 inhibitor, INX-315. Furthermore, CDK2 deletion reverses the G2/M block induced by CDK2 inhibitors and restores cell proliferation. Complementary drug screens define multiple means to cooperate with CDK2 inhibition beyond G1/S. These include the depletion of mitotic regulators as well as CDK4/6 inhibitors cooperate with CDK2 inhibition in multiple phases of the cell cycle. Overall, this study underscores two fundamentally distinct features of response to CDK2 inhibitors that are conditioned by tumor context and could serve as the basis for differential therapeutic strategies in a wide range of cancers.

Mammalian cell cycle machinery has been widely studied and typically viewed as a linear pathway that involves the activation of multiple cyclin dependent kinases (CDKs) that regulate distinct phases of the cell cycle[1,2]. The activation of CDK4 and CDK6 kinases via the binding of D-type cyclins in response to mitogenic signals canonically initiates phosphorylation of the retinoblastoma (RB) tumor suppressor protein, relieving its inhibitory effect on the E2F transcription factor[3,4]. Subsequently, E2F stimulates CDK2 activity via induction of cyclins and other factors, which contribute to RB hyperphosphorylation and leads to the sustained expression of many genes essential for further cell cycle progression, including DNA replication and mitosis[5,6]. Therapeutic intervention at the level of CDK4/6 inhibition has been successful for the treatment of hormone receptor-positive, human epidermal growth factor receptor 2-negative (HR + /HER2-) breast cancer[7–9]. However, the limited therapeutic benefits of CDK4/6 inhibitors beyond HR + / HER2- breast cancer models suggest context-selective cell cycle control with varying dependence on CDK4/6[10–12].

[1]Department of Molecular and Cellular Biology, Roswell Park Comprehensive Cancer Center, Buffalo, NY, USA. [2]Department of Chemistry and Biochemistry, University of California Santa Cruz, Santa Cruz, CA, USA. [3]Incyclix Bio, Durham, NC, USA. [4]Department of Pharmacology and Therapeutics, Roswell Park Comprehensive Cancer Center, Buffalo, NY, USA. ✉e-mail: Agnieszka.Witkiewicz@RoswellPark.org; Erik.Knudsen@Roswellpark.org

It is emerging that cancer cell cycles are highly heterogenous across different tumor types, resulting in exclusive addiction to distinct CDKs[13–15]. For example, in RB-deficient tumor models, the cell cycle is uncoupled from CDK4/6 kinases and D-type cyclins, rendering these proteins dispensable for cell cycle progression[16]. Consequently, RB-deficient tumor models exhibit resistance to pharmacological inhibitors targeting CDK4/6, with these findings validated in multiple preclinical and clinical studies[17–20]. Previous findings have provided mechanistic evidence, demonstrating that in the RB-deficient setting, pharmacological inhibition of CDK4/6 can be circumvented by sustained CDK2 activation that drives cell cycle[16,21]. Multiple studies have underscored the compensatory role of CDK2 activity in overcoming the response to CDK4/6 inhibitors, particularly with overexpression of cyclin E1[21–23]. Thus, aberrant activation of CDK2 is a critical determinant of resistance to CDK4/6 inhibition. Owing to its potential role in regulating cancer cell cycle progression and driving therapeutic resistance to CDK4/6 inhibitors, CDK2 has been considered as an important target for therapeutic intervention[24]. However, the structural similarity in the catalytic domains of CDK2 and CDK1 has posed challenges for drug development[25].

Unlike CDK4/6 which is viewed as generally having a restricted role in G1/S control, the role of CDK2 is more complex. CDK2 phosphorylates a broad range of substrates across various phases of cell cycle beyond G1/S transition[26]. It has been widely viewed that CDK2 in complex with cyclin E1 regulates the G1 to S-phase transition via RB hyperphosphorylation[27,28]. Moreover, CDK2 is also believed to play a critical role in regulating DNA replication and in activating CDK1 thereby allowing mitotic entry[3,29]. Therefore, there is a critical need to explore the pharmacological complexities of CDK2 inhibitors across different tumor models, considering the distinct roles of CDK2 in driving cell cycle progression. To date, understanding the cellular response to pharmacological CDK2 inhibition is limited. Current clinical studies with CDK2 inhibitors mainly focus on cyclin E1 amplified tumors (e.g. ovarian and endometrial) as well as tumors that have progressed with CDK4/6 inhibitors. However, the functional determinants of response/resistance remain poorly understood.

Here we show that cancer context tunes the response to different CDK2 inhibitors. Through different unbiased approaches we identify biomarkers of exceptional response and resistance, which could be further augmented through combinatorial strategies. This work provides the groundwork for expanded clinical implementation of these therapeutic agents.

## Results

### Differential requirements of CDK2 in tumor models

It is progressively accepted that different cancer models exhibit varying dependencies to CDKs and cyclins that intricately regulate their cell cycle progression[13,30]. Utilizing DEPMAP data (23Q4, https://depmap.org/portal/), we employed the CHRONOS score, which is a measure of cell fitness following gene deletion to cluster cancer cell lines ($n = 1100$) based on their dependencies to cyclins and CDKs that regulate the G1 to S phase transition[31]. This analysis segregated cell lines into distinct clusters demonstrating specific dependencies on various cyclin and CDK genes (Fig. 1A). Despite the broad role of CDK2 in the cell cycle, only cluster 3, largely comprising of ovarian and endometrial cancer cell lines, exhibited exclusive vulnerability to *CDK2* and *CCNE1* depletion (Fig. 1A–C). However, loss of *CCNA2*, a key binding partner of CDK2, impacted on the fitness of nearly all the cell lines, underscoring its largely essential function in cell cycle (Fig. 1A, C). In cluster 2, which is enriched for ER+ and/or HER2+ breast cancer and Ewing sarcoma, many cell lines that were sensitive to *CDK4* loss were less affected by *CDK2* depletion, suggesting a reciprocal relationship in the requirements of these kinases (Fig. 1B, C and Supp Fig. S1A). Finally, clusters 5 and 6 encompass a large number

of cell lines whose cell cycle is neither dependent on *CDK4* nor *CDK2*, suggesting that depletion of a single G1/S regulatory kinase is not sufficient to induce cell cycle arrest (Fig. 1B & Supp Fig. S1B).

Consistent with the observations from DepMap, KURAMOCHI, an ovarian cancer cell line, and MB157, a TNBC cell line, both from cluster 3, displayed a robust growth arrest following RNAi-mediated *CDK2* depletion, while *CDK4* depletion showed no effect (Fig. 1D). In contrast, in an HR + /HER2- breast cancer line, MCF7, which belongs to cluster 2, *CDK4* depletion induced a more significant growth arrest (Fig. 1D). Unlike these models, a TNBC cell line, HCC1806, and a patient derived PDAC model, 3226, were not particularly sensitive to either *CDK4* or *CDK2* depletion, (Fig. 1D, Supp Fig. S1C). Biochemical analysis further confirmed that in KURAMOCHI and MB157 cell lines, *CDK2* depletion inhibited the expression of cell cycle regulatory proteins, cyclin A and cyclin B1, which resulted in the inhibition of cell proliferation, while *CDK4* depletion had no impact on these proteins (Fig. 1E). Similarly, in CDK4-addicted HR + /HER2- breast cancer models (MCF7 and CAMA1), the impact on cell cycle proteins following *CDK4* depletion was more pronounced than *CDK2* depletion (Fig. 1E). Finally, the expression of cyclin A and cyclin B1 remained intact in 3226 cells following the depletion of *CDK4* or *CDK2* (Fig. 1E). These data illustrate a complex hierarchy regarding the exceptional dependence on a single kinase, either CDK2 or CDK4. Further analysis revealed diverse responses to *CCNE1* depletion among different models, correlating with the susceptibility to CDK2 loss (Fig. 1F). For instance, the CDK2-sensitive models, MB157 and KURAMOCHI underwent arrest following *CCNE1* depletion while the CDK2-independent cell lines like MCF7 and 3226 were resistant (Fig. 1F). However, *CCNA2* depletion suppressed cell proliferation across different cell lines regardless of their specific vulnerabilities to CDK2 depletion (Fig. 1F, Supp Fig. S1D). The growth arrest following cyclin A depletion in MB157, MCF7 and 3226 cells was associated with an increase in 4N DNA content (Fig. 1G, Supp Fig. S1E). This is distinct from cyclin E1 depletion, which induced cell cycle arrest at the G1 to S phase transition in MB157 cells with a modest effect in MCF7 and 3226 cells (Fig. 1G, Supp Fig. S1E). To define the nature of the 4N arrest we performed bi-variate flow cytometry analysis that determines BrdU incorporation along with DNA content. Although *CCNA2* depletion increased the 4N population in MB157 cells, the cells were unable to enter S-phase to initiate DNA replication as indicated by a pronounced inhibition in BrdU incorporation (Fig. 1H). However, in MCF7 cells, the BrdU profile revealed that the cells underwent DNA replication and arrested at late S-phase or G2 phase with 4N DNA content (Fig. 1H). Biochemical data further confirmed that in the CDK2-addicted models (KURAMOCHI and MB157) *CCNA2* depletion inhibited RB phosphorylation and led to downregulation of cyclin B1, which are the markers for G1 arrest (Fig. 1I). Therefore, CCNA2 depletion in CDK2 addicted models induce a 4N pseudo G1-arrest. This phenomenon was not observed in the MCF7 cells suggesting the cell cycle regulatory functions of cyclin A are distinct in different cell lines (Fig. 1H). Moreover, in MB157 cells *CDK1* depletion induces a 4N block, however, this arrest was associated with enhanced RB phosphorylation and upregulation of Cyclin A and B1, which is different than the effect of *CCNA2* knockdown indicating the distinct regulatory roles of these proteins in cell cycle (Supp Figs. S1F, G). Collectively these data underscore the context-selective roles of CDK and associated cyclin genes in regulating G1/S transition versus other phases of the cell cycle. Moreover, based on the differential dependencies to CDK2, only a small fraction of tumors would be exceptionally responsive to a selective pharmacological CDK2 inhibitor.

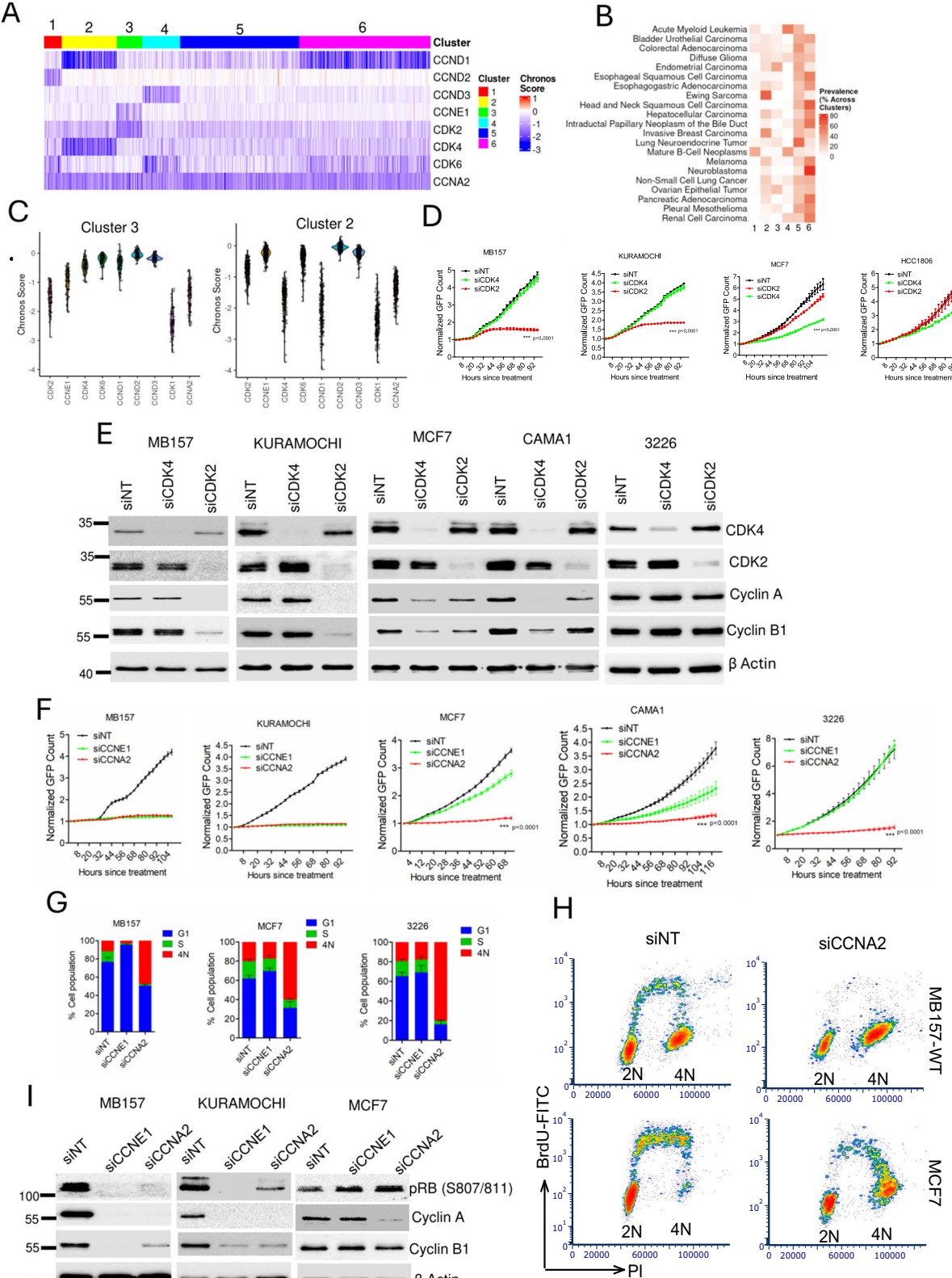

### Defining the exceptional response to pharmacological CDK2 inhibition

INX-315, a selective CDK2 inhibitor, induced cell cycle arrest at distinct phases of cell cycle in the CDK2-addicted MB157 cell line, depending on the drug dosage (Fig. 2A). In the lower concentration range (50 to 100 nM), INX-315 inhibited the G1 to S phase transition, while at higher concentrations (250 to 1000 nM), cell population with 4N DNA content

gradually increased (Fig. 2A). This distinct cell cycle arrest depending on the dosing corroborates with the differential binding affinity of INX-315 for cyclin E1/CDK2 complex, which is approximately 10-fold higher than that for the Cyclin A/CDK2 complex as described in the previous study (Supp Fig. S2A)[32]. Bivariate flow cytometry analysis revealed that at the higher concentration range (250 & 500 nM), INX-315 inhibited the entry into S-phase to initiate DNA replication, indicating a pseudo-

**Fig. 1 | Heterogeneity in the requirements of different CDKs and cyclins. A** Heat map depicting the distinct clustering of multiple tumor cell lines based on their dependencies to the indicated CDKs and cyclins. The dependency was measured based on the CHRONOS scores from DEPMAP dataset. **B** The enrichment of different tumor types within each cluster. **C** Violin plots illustrating the distribution of cell lines from cluster 3 and cluster 2 based on their dependencies to indicated genes. **D** Live cell imaging to monitor the proliferation of the indicated cell lines following RNAi-mediated depletion of *CDK4* and *CDK2*. Representative cell lines were selected from clusters 2, 3 and 6. Error bars indicate mean and SD from $n = 3$ technical replicates. The experiments were done as 3 biological replicates. *** represents *p* value < 0.0001 as determined by 2-way ANOVA comparing siCDK4 and siCDK2. **E** Biochemical analysis to interrogate the impact of *CDK4* and *CDK2* depletion on cell cycle proteins on the indicated cell lines. **F** Differential effect

of *CCNE1* and *CCNA2* depletion on the proliferation of indicated cell lines. Error bars were determined based on mean and SD from $n = 4$ technical replicates. The experiments were repeated 3 independent times. *** indicates *p* value < 0.0001 as determined by 2-way ANOVA, comparing *CCNE1* and *CCNA2* depletion. **G** Cell cycle analysis on MCF7, MB157 and 3226 cell lines following *CCNE1* and *CCNA2* depletion. The cell population at each phase was quantified and graphed. Error bars represent mean and SD from $n = 3$ biological replicates. **H** Bivariate flow cytometry analysis to determine the BrdU incorporation in MB157 and MCF7 cells following the *CCNA2* depletion. **I** Western blotting on MB157, KURAMOCHI and MCF7 following the deletion of CCNE1 and *CCNA2* to determine the effect on cell cycle proteins. Western blotting was done at two independent times (**E & I**). Source data are provided as a source data file.

---

G1 arrest with 4N DNA content, similar to *CCNA2* depletion (Fig. 2B). This phenomenon was biochemically confirmed based on RB dephosphorylation and downregulation of E2F target genes, such as cyclin A and cyclin B1, which are markers of G1-arrest (Fig. 2C). Upregulation of cyclin E1 is likely an indication of CDK2 inhibition, as cyclin E1 degradation is mediated in part by CDK2-mediated phosphorylation (Fig. 2C)[33]. The INX-315-mediated pseudo-G1 arrest was visualized in real time using PIP-FUCCI, a live cell imaging reporter system that tracks the transition between cell cycle phases based on fluorescent signals[34]. The PIP-FUCCI consists of mVenus-tagged Cdt1, marking G1-phase, mCherry-tagged Geminin, defines S-phase and the co-expression of both fluorescent- proteins define G2/M phase. We stably expressed PIP-FUCCI in MB157 cells and treated them with INX-315 (500 nM). Single cell tracking revealed that INX-315 treatment allowed transition of cells from S to G2/M phase but were unable to complete mitosis. These cells re-entered the G1 phase and remained arrested (Fig. 2D & Supp Fig. S2B). Moreover, cells being at the G1-phase were unable to enter S-phase for DNA replication (Supp Fig. S2B). Quantitatively, we observed a significant increase in the mVenus-positive cells with a corresponding decrease in the mCherry positive cells, confirming G1-arrest (Fig. 2D). Moreover, the growth arrest in MB157 cells following INX-315 treatment was associated with an increase in cell size, and increased senescence- associated β-galactosidase staining (Fig. S2C & Supp Fig. S2D). The accumulation of cells with 4N DNA content is not reflective of DNA damage as no γH2AX Foci formation was observed in the presence of INX-315 (Supp Fig. S2E). As a positive control we used a CHKi (CHIR-124), which increased γH2AX[32].

Given the heterogeneous response to CCNE1 depletion among tumor types, we examined the impact of INX-315 at low concentrations across a panel of cell lines. At concentrations as low as 31 nM, INX-315 inhibited the proliferation of MB157 and KURAMOCHI cells (Fig. 2E, Supp Fig. S2F). However, MCF7, 1222, 3226, MIA PaCa-2 and HCC1806 cells were not impacted by INX-315 at these doses, indicating a context-dependent cellular response (Fig. 2E).

To elucidate the effect of CDK2 inhibition on gene expression we performed transcriptome analysis in the sensitive model MB157 and in two relatively resistant models, MCF7 and HCC1806. Consistent with their intrinsic sensitivity to INX-315, the MB157 cells displayed a significant modulation of gene expression compared to the modest effect observed in MCF7 and HCC1806 cells (Fig. S3A, Supp Data 1). GSEA analysis revealed that the top pathway that was significantly impacted in MB157 cells by INX-315 was the E2F targets pathway (Fig. 2F). To interrogate whether the INX-315-mediated downregulation of E2F target genes in MB157 cells occurs via RB activation, we developed an isogeneic cell line, MB157-RB-del through Cas9-mediated RB deletion. RB loss in MB157 cells uncoupled dependence on cyclin E1/CDK2 complex, which was evident by the sustained cell proliferation despite the depletion of *CCNE1* or *CDK2* (Supp Fig. S3B). Consistent with this data, treatment with INX-315 at low concentrations had modest impact on the proliferation of MB157-RB-del cells (Fig. 2G). Transcriptome analysis further corroborated this resistance phenomenon, revealing very few

differentially expressed genes following INX-315 treatment (Supp Fig. S3C, Supp Data 1). The INX-315 mediated upregulation of cyclin E1 was observed in both MB157-WT and RB-del cells indicating that the regulation of this protein is independent of RB activation, while the downregulation of cyclin A is dependent on RB activation (Supp Fig. S3D).

To further evaluate therapeutic response in the context of exceptional dependence, orthotopic MB157 xenografts were developed. Consistent with the cell culture data, INX-315 significantly inhibited tumor growth of MB157 xenografts and resulted in disease control (Fig. 2H, Supp Fig. S3E). Histological analysis revealed that INX-315 treatment resulted in tumor cells with larger nuclei as compared to that of the vehicle-treated tumor (Fig. 2I). To define the molecular impact of INX-315 in vivo, multispectral immunofluorescent staining was carried out on the tumor tissue (Fig. 2J). Similar to the results observed from cell culture analysis, INX-315 significantly inhibited the phosphorylation of RB and upregulated cyclin E1 expression in the xenografts (Fig. 2K). Further biochemical analysis confirmed that INX-315 inhibited RB phosphorylation that resulted in reduced expression of cyclin A and cyclin B1 (Supp Fig. S3F). Collectively, these findings demonstrate an exceptional response to INX-315 in CDK2-addicted models via induction of a pronounced RB-dependent G1 arrest.

### P16INK4A and cyclin E1 are biomarkers of the potent response to CDK2 inhibition

To determine features associated with the exceptional sensitivity to CDK2 inhibition, we evaluated the differential gene expression that defines the CDK2 dependent cluster as compared to the CDK4-dependent cluster (Cluster 2) and CDK4/CDK2 independent cluster (Cluster 6). Our data revealed that the expressions of *CCNE1* and *CDKN2A*, which encodes P16INK4A, an endogenous inhibitor of CDK4 and 6 kinases, were significantly higher in the CDK2-addicted cluster 3 as compared to cluster 2 and 6 (Fig. 3A). Comparing two *CCNE1*-amplified models (MB157 & HCC1806) illustrated that the expression of P16INK4A was associated with reliance on CDK2 and cyclin E1 (Fig. 3B). Exogenous overexpression of P16INK4A in HCC1806 cells potentiated the impact of *CCNE1* depletion and resulted in RB activation (Supp Fig. S4A). Furthermore, the pharmacological inhibition of CDK2 using INX-315 in P16INK4A-overexpressing HCC1806 cells elicited a pronounced reduction of RB phosphorylation and cyclin A downregulation (Fig. 3C). Consistent with P16INK4A overexpression, concurrent depletion of CDK4 and 6 kinases in HCC1806 cells resulted in G1 arrest following INX-315 treatment, which is associated with RB dephosphorylation and suppression of cell cycle proteins (Supp Fig. S4B, C). To further mimic the CDK2/cyclin E1-addicted state, P16INK4A and cyclin E1 were co-expressed in MCF7 cells. Ectopic overexpression of P16INK4A induced a durable cell cycle arrest, which could be partially alleviated by overexpressing cyclin E1 (Supp Figs. S4D, E).

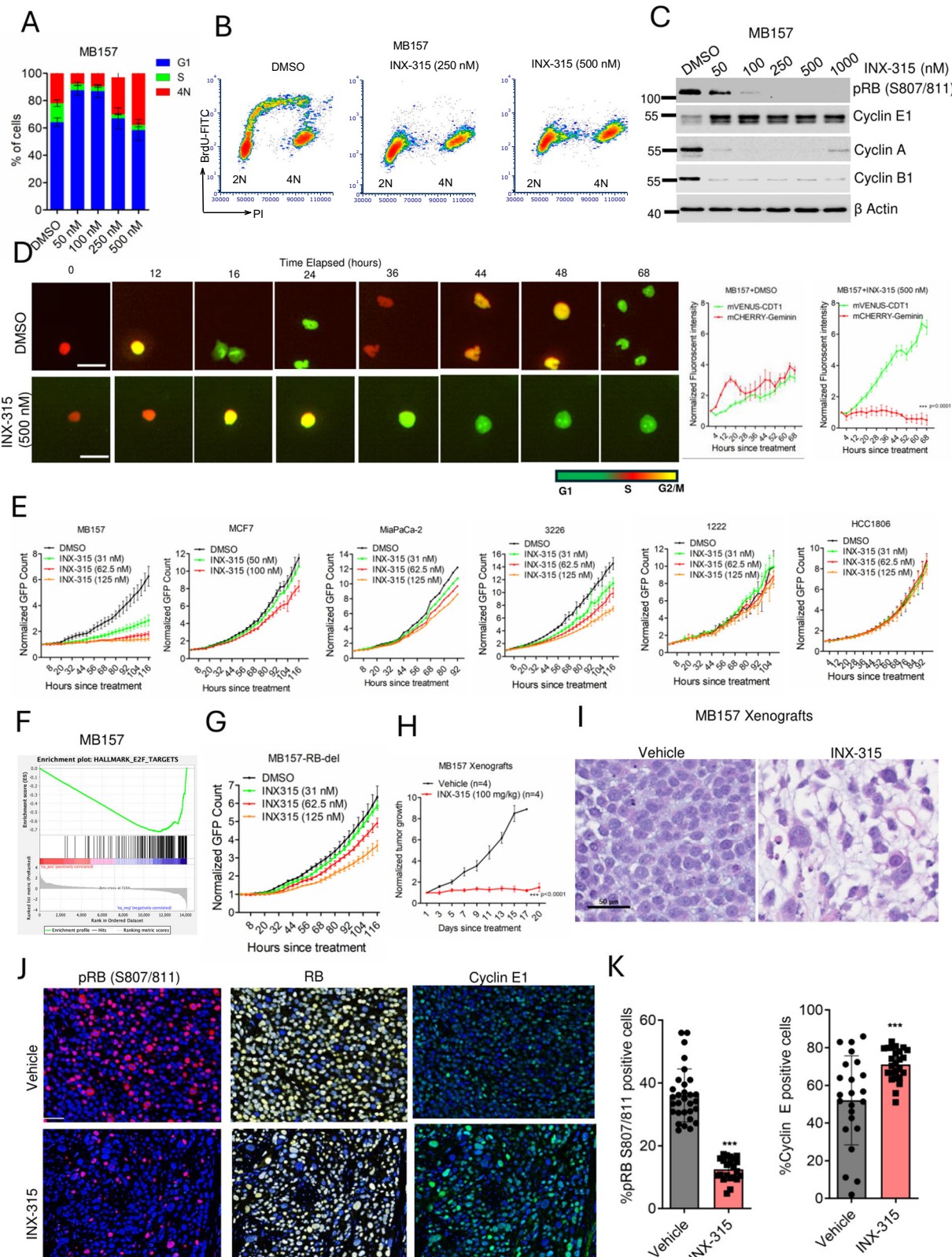

Notably, the addition of INX-315 following the overexpression of cyclin E1 and P16INK4A in MCF7 cells resulted in an enhanced cycle arrest (Supp Fig. S4D, E). Conversely, depleting *CDKN2A* in one of the CDK2 addicted models (KURAMOCHI) using gene specific RNAi, significantly mitigated the INX-315-mediated cytostatic response (Fig. 3D). Biochemical analysis confirmed that following the *CDKN2A* depletion, cells retained RB phosphorylation and

cyclin A expression despite the presence of INX-315 (Fig. 3E). As a complementary approach, exogenous overexpression of CDK4 was employed to titrate out P16INK4A in the CDK2 addicted models, KURAMOCHI and MB157 cells, which conferred resistance to INX-315 (Fig. 3F, G). Collectively these findings suggest that the co-expression of both p16INK4A and cyclin E1 are critical for response to CDK2 inhibition.

**Fig. 2 | Pharmacology of INX-315, a CDK2 inhibitor. A** Cell cycle analysis on MB157 cells following the treatment with different concentrations of INX-315 for up to 48 h. Error bars were calculated based on mean and SD from five biological replicates (*n* = 5). **B** Bivariate flow cytometry analysis to determine the BrdU incorporation in MB157 following INX-315 treatment for 48 h. **C** Biochemical analysis on the indicated proteins from MB157 cells after 48-hour exposure with INX-315. Western blotting was done at two independent times. **D** Representative images of MB157 cells expressing PIP-FUCCI to demonstrate the transition of different cell cycle phases based on the mVENUS and mCHERRY fluorescent signals over the indicated period of time in the presence and absence of INX-315 (500 nM). Scale bar represents 100 microns Graphs represent the relative change in the fluorescent intensities determined in real time using IncuCyte. Error bars represent mean and SEM from *n* = 4 technical replicates. Experiment was carried out at 3 biological replicates. Statistical significance was determined based on 2-way ANOVA (*** represents *p* value < 0.0001, comparing mVenus-CDT1 and mCherry-Geminin) (**E**) Live cell imaging to monitor the proliferation of the indicated cell lines following the treatment with different concentrations of INX-315 until the indicated time points. The data points for MiaPaCa-2 cells represent mean from *n* = 4 technical replicates. **F** GSEA analysis identified a significant enrichment of E2F targets, which is downregulated following the treatment with INX-315 (100 nM). **G** Cell proliferation of MB157-RB-del cell line following the treatment with increasing

concentrations of INX-315 until the indicated time points. Error bars were determined based on mean, and SD from triplicates. Experiment was repeated at 3 independent times for (**E & G**). **H** Relative tumor growth rate of MB157 xenografts that were developed by orthotopically injecting cells into mammary fat pad in NSG mice. INX-315 (100 mg/kg) was administered orally once/day for the indicated number of days. Error bars represent mean and SEM for vehicle-treated (*n* = 4 mice) and INX-315-treated (*n* = 4 mice) groups. *** represents *p* value < 0.0001 as determined by 2-way ANOVA, comparing the vehicle- and INX-315-treated groups. **I** Representative images of the (**H** & **E**) staining on the tumor tissues that were treated with vehicle and INX-315. Scale bar 50 microns. **J** Representative MIF staining images to demonstrate the effect of INX-315 on RB phosphorylation and cyclin E1 expression. The scale bar represents 50 microns. **K** The column graph illustrates mean and SD based on the fraction of pRB positive cells from vehicle (*n* = 10 technical replicates from *n* = 3 mice) and INX-315-treated groups (*n* = 8 technical replicates, from *n* = 3 mice). The mean and SD were determined based on cyclin E1 positive cells from the vehicle (*n* = 8 technical replicates; *n* = 3 mice) and INX-315-treated (*n* = 8 technical replicates, *n* = 3 mice) groups. Statistical analysis was carried out using two-tailed unpaired student t-test (***p* < 0.0001, comparing the vehicle- and INX-315-treated groups). Source data is provided as a source data file.

## Biomarkers of response in breast cancer patients

To determine the extent to which the co-expression of high cyclin E1 and P16INK4A occur in human tumors we employed multispectral immunofluorescent staining on tissue microarrays that comprised 197 TNBC cases[35,36]. While the high expression of P16INK4A is associated with RB loss, there were 21 tumors (10% cases) with high P16INK4A and cyclin E1 protein expression that were RB positive (Fig. 4A, B)[37]. By correlation and clustering analysis, cyclin E1 and P16INK4A were co-expressed with proliferation associated markers, cyclin A and Geminin. (Fig. 4C). Gene expression analysis of TNBC subtype from METABRIC dataset further revealed a significant positive correlation between the expression levels of *CDKN2A* and *CCNE1* in individual tumors, (Fig. 4D). Prior studies have indicated that high expressions of P16INK4A or cyclin E1 are associated with resistance to CDK4/6 inhibitors[30,38–40]. Therefore, we interrogated the *CDKN2A* and *CCNE1* expression in of ER+ tumors express, of which few tumors expressed high levels of both the markers as compared with TNBC (Fig. 4D). This finding aligns with the better efficacy of CDK4/6 inhibitors in ER+ breast cancer as compared to TNBC. High expression of *CDKN2A* and *CCNE1* was observed predominantly in basal-like ER+ tumors, which is consistent with clinical findings that basal-like ER+ tumors have minimal therapeutic benefit from CDK4/6 inhibitors (Fig. 4E)[41]. Analysis from PALOMA-2 and PEARL trials indicated a positive correlation between *CCNE1* and *CDKN2A*, however, only a few basal like tumors were represented and did not form a distinct cluster of *CCNE1/CDKN2A* high population[42,43] (Fig. 4F). We also analyzed our own ER+ breast cancer patient cohort (NCT04526587), comparing the expression of *CCNE1* and *CDKN2A* before and following progression on CDK4/6 inhibitor-based treatment[44]. These results indicated a significant increase in *CCNE1* expression postprogression (Fig. 4G). The overall levels of *CDKN2A* were unchanged: however, there were several patients with elevated *CDKN2A* and high *CCNE1* expressions which exhibited very short progression-free survival (Fig. 4G). These analyzes support the potential predictive value of P16INK4A and cyclin E1 for CDK-inhibitor sensitivity in the clinical setting.

## Mechanistic Impact of CDK2 inhibition beyond G1/S

Considering the impact of *CCNA2* depletion on cell proliferation across different cell lines, we investigated the response to higher doses of INX-315. Notably, the models that were insensitive to

CDK2 gene depletion, such as MCF7, 3226, 1222, MIA PaCa-2 and HCC1806 demonstrated sensitivity to INX-315 at higher concentrations (Fig. 5A). These cell lines also responded similarly to another CDK2-selective inhibitor, PF-07104091, indicating that the observed effects are a feature of CDK2 inhibitors rather than a compound-selective effect (Supp Fig. S5A). The growth arrest induced by INX-315 was associated with accumulation of cells with 4N DNA content, which is similar to the response to *CCNA2* depletion (Supp Fig. S5B). BrdU incorporation based on bivariate flow cytometry analysis suggests that the cell cycle arrest occurs after DNA replication indicating a late S-phase or G2/M arrest (Fig. 5B). Biochemical analysis confirmed that the INX-315-mediated G2/M arrest is associated with enhanced RB phosphorylation, increased inhibitory phosphorylation of CDK1 (Y15) and accumulation of cyclin B1 (Fig. 5C). This observation is distinct from the response observed in the exceptionally sensitive model, MB157, where RB activation prevented CDK1 phosphorylation and suppressed cyclin B1 expression (Fig. 5D). However, in the isogenic MB157-RB-del cells, the impact on cell proliferation is mediated through the activation of G2/M checkpoint (Fig. 5D & Supp Fig. S5C). Interestingly, the upregulation of cyclin E1 by INX-315 was observed in all the cell lines tested, indicating this response is a direct action of CDK2 inhibition (Fig. 5C, D).

We interrogated whether the enhanced expressions of different cyclins in the presence of INX-315 alters the complex formation with the CDKs using co-immunoprecipitation assays. In actively proliferating MCF7 and MiaPaCa-2 cells, CDK2 functions to consistently bind and retain cyclin E1 and cyclin A, as both proteins were present at low levels in the flowthrough (Fig. 5E). Following INX-315 treatment, although the expression of cyclin E1 and cyclin A were increased, the excess proteins predominantly bound to CDK2, with minimal unbound proteins detected in the flowthrough (Fig. 5E). Therefore, INX-315-mediated growth arrest is associated with a significant increase in CDK2 complex formation with cyclin E1 and cyclin A in both MCF7 and MiaPaCa-2 cells (Fig. 5F). Despite the increase in expression of cyclin B1, the fraction of it in complex with CDK2 and CDK1 remains very low and the impact of INX-315 on those complexes is very modest (Fig. 5F & Supp Fig. S5D). In order to evaluate whether the increase in CDK2-cyclin A complex leads to increased stability we used a previously described biolayer interferometry (BLI) assay to monitor the association of purified CDK2 with cyclin A (Supp Fig. S5E)[45]. The addition of INX-315 to CDK2 significantly increased the stability of cyclin A by decreasing the cyclin dissociation rate constant which is consistent with the behavior of other Type I

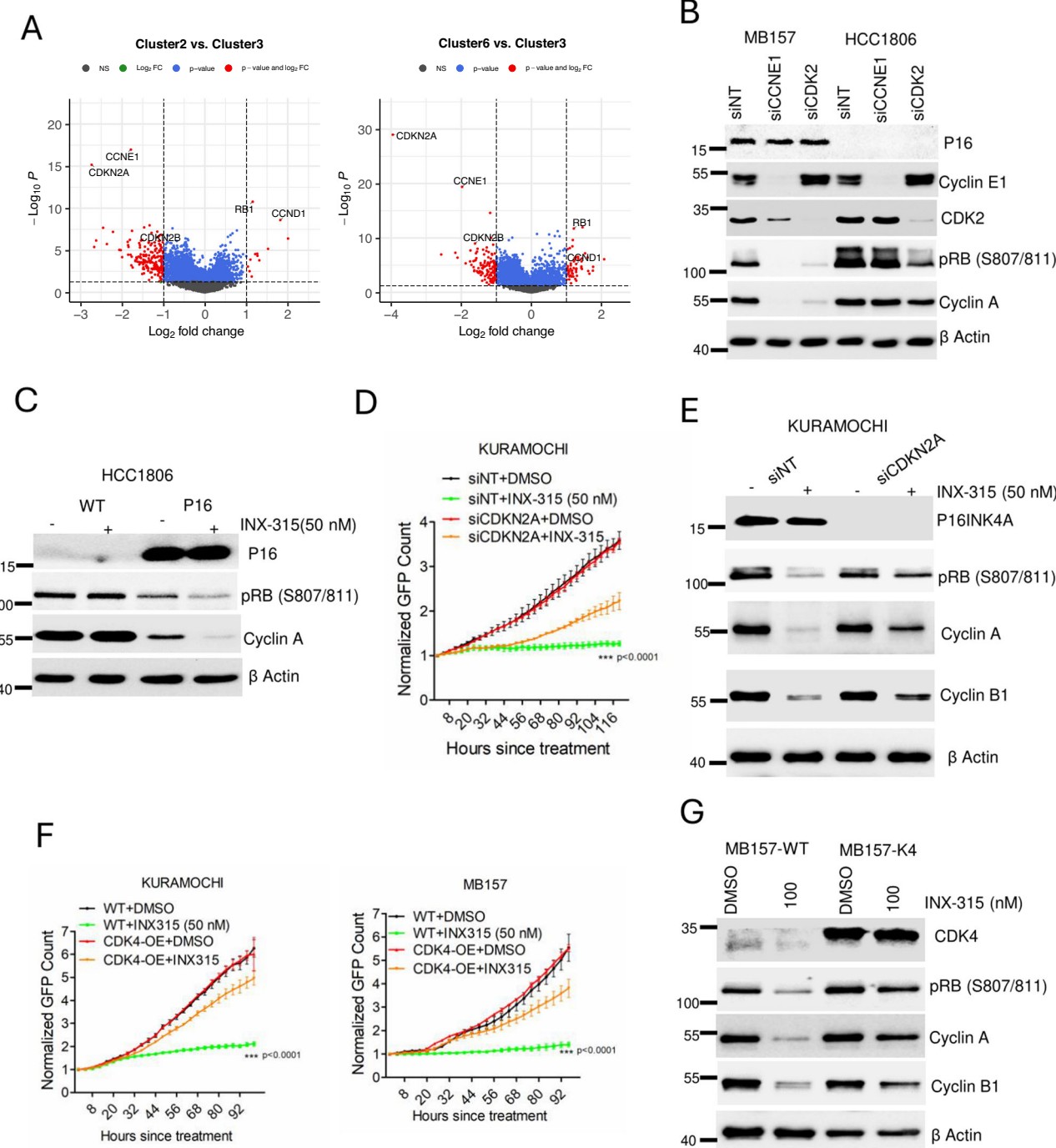

**Fig. 3 | Biomarkers to predict sensitivity to INX-315. A** Volcano plot illustrating the differential gene expression comparing cluster 3 (CDK2 addicted) versus cluster 2 (CDK4 addicted) and cluster 3 versus cluster 6 as mentioned in Fig. 1. The *X* axis represents log$_2$fold change and *Y* axis represents -log$_{10}$P. The *P* value was calculated based on two-tailed unpaired student t-test (**B**) Differential effect of *CCNE1* and *CDK2* depletion on cell cycle proteins as determined by western blotting in MB157 and HCC1806 cell lines. Western blotting was done at two independent times (**C**) Impact of ectopic overexpression of P16INK4A in HCC1806 cells on the cellular response to INX-315 (nM) following 48 h treatment as determined by western blotting. Western blotting was done at two independent times (**D**) Live cell imaging to monitor the impact of *CDKN2A* depletion in KURAMOCHI cells following the treatment with INX-315 (50 nM). Error bars were determined based on mean and SD from triplicates. The experiment was done at three independent times (*n* = 3).

(*** indicates *p* value < 0.0001 as determined by 2-way ANOVA comparing siNT +INX-315 and siCDKN2A+INX-315). **E** Biochemical analysis to determine impact of *CDKN2A* depletion on the cell cycle proteins in the absence and presence of INX-315 (50 nM). **F** Live cell imaging to monitor the growth of KURAMOCHI-WT, KURAMOCHI-K4, MB157-WT and MB157-K4 cells in the absence and presence of INX-315. Error bars were determined from mean and SD from triplicates. Experiments were done at n = 3 independent times. *** indicates *p* value < 0.0001 as determined by 2 way-ANOVA comparing WT + INX-315 and CDK4-OE + INX-315. **G** Differential effect of INX-315 on the cell cycle proteins from MB157-WT and its isogeneic counterpart, MB157-K4, which harbors ectopic overexpression of CDK4 after 48-hour exposure. Western blotting was done at two independent times (**B, C, E & G**). Source data are provided as a source data file.

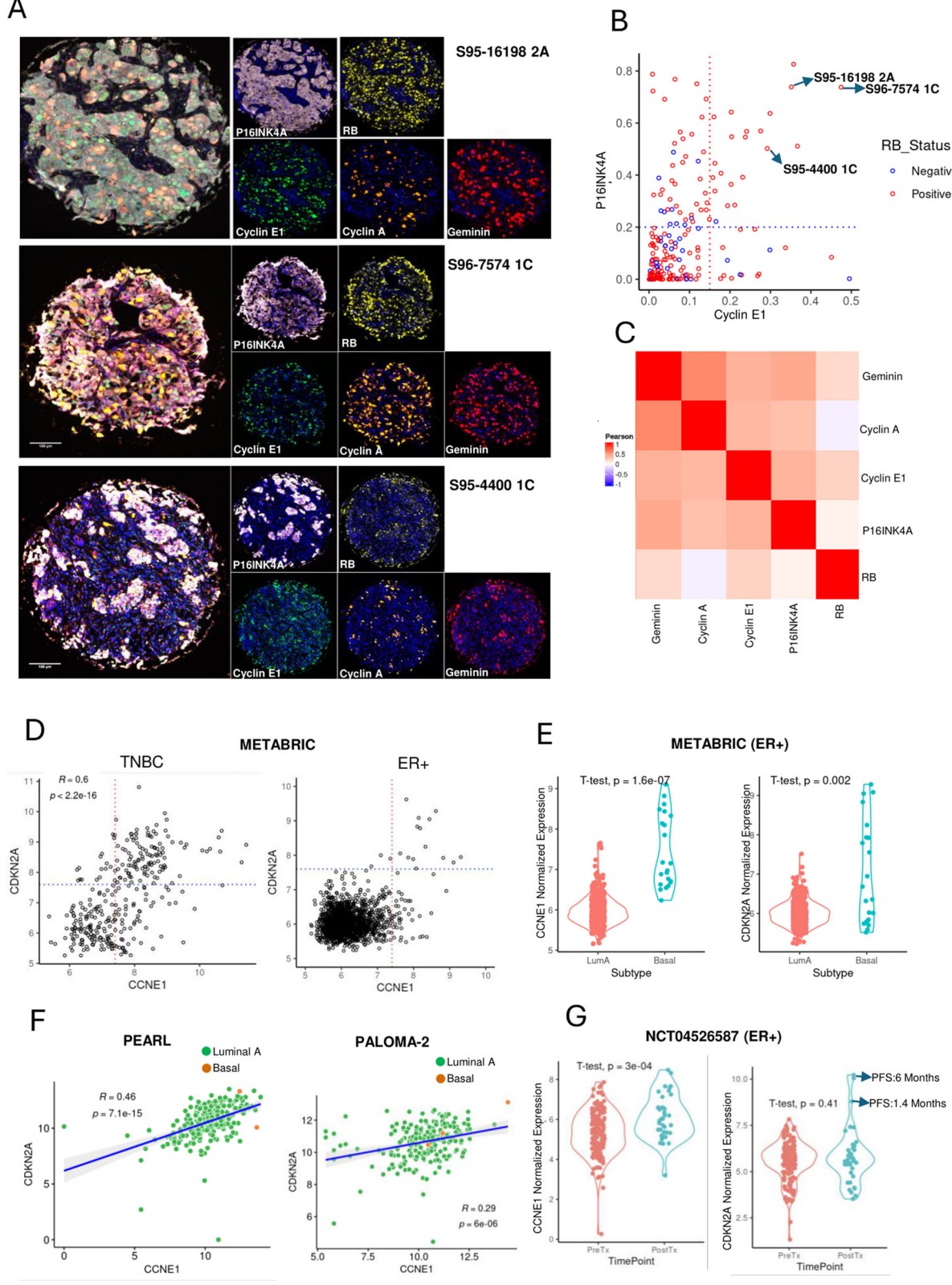

kinase inhibitors (Supp Fig. S5E)[45]. To assess the specificity of INX-315, we compared the inhibition of catalytic activities of CDK2 and CDK1 using in vitro kinase reactions. Based on the phosphorylation of C-terminal RB peptide substrate, INX-315 exhibited significantly higher selectivity for CDK2 vs CDK1 (Fig. 5G, H). This observation aligns with the previously published data, indicating that INX-315 exhibits greater affinity for CDK2 than for CDK1 (Supp Figs. S2A, S5E)[32]. Additionally, we

compared the cellular response between INX-315 and a CDK1-selective inhibitor, RO-3306[46]. While both the drugs induced a comparable 4N block at the indicated doses (Supp Fig. S6A), RO-3306 more potently induced the phosphorylation of Histone H3 (S10) (pHH3) as compared to INX-315 in 3226 cells, which is a marker for mitotic arrest (Supp Fig. S6B). These findings demonstrate a distinct cellular outcome between INX-315 and CDK1 inhibition.

**Fig. 4 | Co-expression of P16INK4A and cyclin E1 in breast cancer patients.**
**A** Representative images based on the multi-spectral immunofluorescence staining on the TMAs. The TMA was stained to detect the expression of P16INK4A, cyclin E1, RB, Cyclin A and Geminin. A total of 197 tissues were stained. The scale bar represents 100 microns. **B** Scatter plot illustrates the correlation between the intensities of cyclin E1 and P16INK4A as determined by multi-spectral immuno-fluorescence (MIF) on the TNBC TMA that comprise 197 tissues. The samples were characterized as RB proficient and RB-deficient. **C** Heatmap depicting the correlation between the fraction of tumor cells expressing the indicated proteins from individual patients based on the MIF staining on the TMA. **D** Pearson's correlation analysis between the gene expression of *CDKN2A* and *CCNE1* from TNBC ($n = 320$) and ER+ ($n = 1506$) breast cancer patients based on the METABRIC datasets. The statistical analysis was determined using the t-distribution method and the $p$ value < 0.0001. **E** Violin plots depicting the differential expression of *CCNE1*

and *CDKN2A* between the luminal ($n = 563$) and basal subtypes ($n = 22$). Statistical analysis was carried out using the two-tailed student t-test. The $p$ value for *CCNE1* expression is <0.0001 and for *CDKN2A* is 0.002. **F** Pearson's correlation analysis between the gene expression of *CDKN2A* and *CCNE1* from ER+ breast cancer patients from PEARL ($n = 258$ patients) and PALOMA-2 ($n = 234$ patients) trials[42,43]. The samples were characterized as luminal and basal subtypes. Statistical analysis was done using the t-distribution method (*** indicates $p$ value < 0.0001). **G** Violin plots depicting the differential expression of *CDKN2A* and *CCNE1* between the CDK4/6 inhibitor pre-treatment ($n = 170$ patients) and post-treatment ($n = 39$ patients). The PFS of the two patients who express high level of *CDKN2A* was illustrated. Statistical analysis was done using the two-tailed student t-test. The $p$ value for CCNE1 expressions is < 0.0001 and the $p$ value for CDKN2A expression is 0.41.

## CDK2 expression mediates response to pharmacological inhibitors

To identify genes that cooperate with INX-315 to mediate growth arrest, a genome wide loss-of-function CRISPR screen was carried out in two cell lines, MIA PaCa-2 and HCC1806, using the TKOV3 library[47,48]. Following infection with the library, the HCC1806 and MIA PaCa-2 cells were grown in media supplemented with vehicle or INX-315 (500 nM) for up to 5 passages. The guides (sgRNAs) that were positively and negatively enriched following drug treatment were determined by the DrugZ algorithm[49]. Our analysis revealed an unexpected observation that CDK2 deletion was the most positively selected event with INX-315 treatment. (Fig. 6A, Supp Data 2). Indeed, the guides targeting *CDK2* were highly enriched following the treatment with INX-315 in both MIA PaCa-2 and HCC1806 cells, demonstrating that *CDK2* deletion confers resistance to INX-315 (Fig. 6B). Further validation using gene-specific siRNA indicated that growth arrest following INX-315 treatment in MCF7 cells and MiaPaCa-2 cells was significantly mitigated following CDK2 depletion (Fig. 6C & Supp Fig. S6C). CDK2 depletion reversed the INX-315-mediated 4N arrest, mitigated the induction of tyrosine phosphorylation of CDK1 and prevented the accumulation of cyclin B1 in MIA PaCa-2 and MCF7 cells (Fig. 6D, E). To validate these observations, we generated isogenic cell lines (MiaPaCa-2-sgCDK2, 3226-sgCDK2 and MCF7 sg-CDK2) that harbor deletion of CDK2 using CRISPR-CAS9 approach. Consistent with RNAi data, CDK2 deletion reduced sensitivity to pharmacological inhibitors, INX-315 and PF-07104091 and reversed the 4N block and CDK1 tyrosine phos-phorylation (Fig. 6F–H & Supp Fig. S6D). The resistance driven by CDK2 deletion was selective for CDK2 inhibitors (PF-07104091 and INX-315), as the impact of RO-3306 on cell proliferation and 4N block remained unaffected (Fig. 6I, J & Supp Fig. S6E). Collectively these data highlight that the expression of CDK2 is an critical mediator for the cytostatic efficacy of CDK2 inhibitors in these models.

## Cell cycle regulators that cooperate with CDK2 inhibitors

Further evaluation of the CRISPR screen unveiled a substantial depletion of sgRNAs following the selection with INX-315 in both HCC1806 ($n = 512$) and MIA PaCa-2 ($n = 520$) cells with NormZ value < -2 (Fig. 7A, Supp Data 2). Based on ENRICHR analysis from HCC1806 and MIA PaCa-2 cells, we identified that the genes that were targeted were involved in cell cycle and regulation of G2/M phase of the cell cycle machinery (Supp Fig. S7A). Notably, the individual guides targeting FOXM1 and CCNB1 were depleted in both HCC1806 and MIA PaCa-2 cells (Supp Fig. S7B). Subsequent validation using gene-specific siRNA showed that depletion of FOXM1 and cyclin B1 significantly augmented the cellular response to CDK2 inhibitors, INX-315 and PF-07104091 in MIA PaCa-2, HCC1806 and 3226 cell lines (Fig. 7B–E). Given their roles as key regulators of CDK1 kinase, we evaluated the impact of *CDK1*

depletion on the cellular response to INX-315[50,51]. Since CDK1 is a pan-essential gene, we lowered the siRNA concentration followed by the treatment with INX-315, which significantly enhanced the anti-proliferative effect of INX-315 in MiaPaCa-2 and HCC1806 cells (Fig. 7F). Depletion of *FOXM1*, *CCNB1* and *CDK1* enhanced the efficacy of INX-315 in accumulation of cells with 4N DNA content in MiaPaCa-2 and HCC1806 cells (Fig. 7G & Supp S7C). Therefore, these data illustrate that genetically impacting the CDK1 pathway enhances the efficacy of CDK2 inhibitors.

## Co-targeting CDK4/6 and CDK2 as therapeutic approach

As a complementary approach to the CRISPR screens, combinatorial drug screens were performed in CDK2-independent cell lines (1222, 3226, and HCC1806), following pretreatment with INX-315. Correlation analysis revealed that targeted therapeutic agents such as MEK and mTOR inhibitors, as well as drugs interfering with cell cycle, such as AURK and CDK inhibitors, emerged as lead agents to inhibit cell pro-liferation across the three cell lines (Fig. 8A & Supp Fig. S8, S9). Pal-bociclib, a CDK4/6 inhibitor, was identified as a common candidate that induced growth arrest in all cell lines tested (Fig. 8B). Further validation in multiple cell lines confirmed that palbociclib in combi-nation with INX-315 cooperated in growth arrest, indicating a potential therapeutic approach (Fig. 8C).

CRISPR/Cas9 screens in MIA PaCa-2 and HCC1806 identified *CCNE1* as the most negatively selected gene upon palbociclib treat-ment, underscoring the therapeutic impact of coordinately targeting CDK2 and CDK4/6 kinases (Fig. 8D, Supp Data 3). Moreover, sgRNAs targeting *RB1* were highly enriched following palbociclib selection (Fig. 8D, Supp Data 3). RNAi-mediated depletion of *CCNE1* and *CDK2* significantly enhanced the efficacy of palbociclib in MIA PaCa-2 cells, thereby inducing a growth arrest in an RB dependent manner (Fig. S10A, B). Bivariate flow cytometry analysis further confirmed a G1 arrest, resulting in the inhibition of BrdU incorporation in MIA PaCa-2 and 1222 cell lines (Supp Fig. S10C). This is consistent with our prior studies that CDK2 activity is a key determinant of response to CDK4/6 inhibitors. Since CDK4 and 6 kinases are exclusively involved in G1-Sphase transition via RB activation, limiting the G2/M regulatory ele-ments like FOXM1, had modest impact on the efficacy of CDK4/6 inhibitors (Supp Fig. S10D).

To define the mechanistic impact of co-targeting CDK4/6 and CDK2, we examined the effect on cell cycle progression. Surprisingly, the combination of palbociclib with INX-315 (Palbo/INX-315) in MIA PaCa-2 and HCC1806 cells did not exclusively arrest the cells at G1 phase as observed with the *CDK2* depletion. Bi-variate flow cytometry analysis revealed that, although the cells entered S-phase, the repli-cation potential was impacted following the combination treatment as indicated by a significant inhibition in BrdU incorporation (Fig. 8E). Furthermore, a synergistic effect was observed between palbociclib and INX-315 based on BrdU incorporation across multiple models tested (Fig. 8F & Supp Fig. S11A). Biochemical analysis indicated that

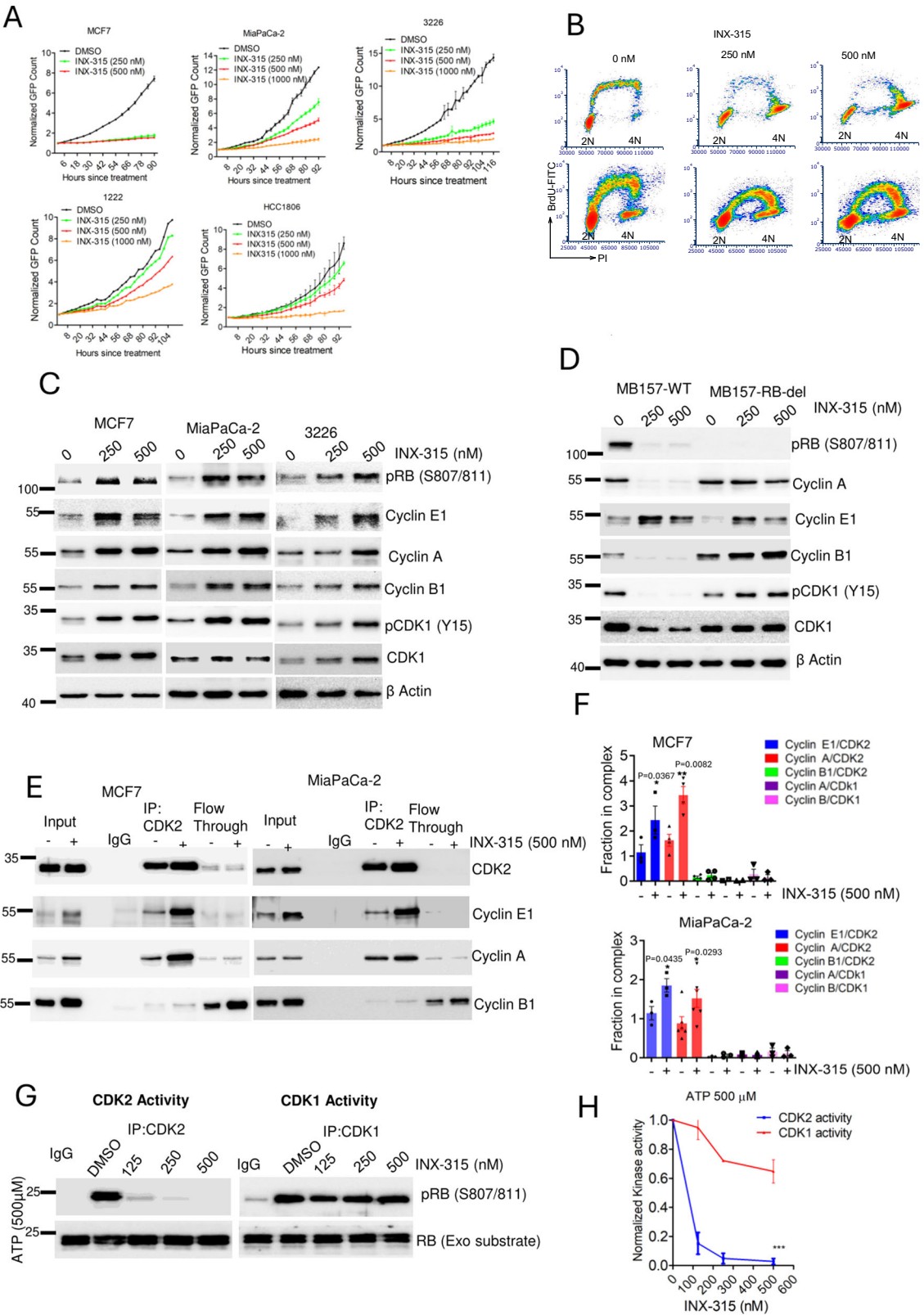

the combination treatment resulted in a cooperative inhibition of RB phosphorylation that further suppressed E2F-regulated proteins such as cyclin A, cyclin B1, and CDK1 (Fig. 8G). The minimal effect of Palbo/INX-315 on cell cycle proteins in the MIA PaCa-2-RB-del cells suggests that the impact of INX-315 on the CDK2 complexes augments the effect of palbociclib in RB dependent manner (Fig. 8H). These data indicate that the concurrent treatment involving palbociclib and INX-315

inhibits cell proliferation, which is associated with a G1 cell cycle arrest and DNA replication collapse.

## In vivo efficacy of CDK2 inhibitors in models of intermediate sensitivity

Since our in vitro data demonstrated that higher doses of CDK2 inhibitor suppresses cell proliferation in CDK2 independent cell lines, we

**Fig. 5 | Cellular response to INX-315. A** Effect of INX-315 on the proliferation of indicated cell lines. Error bars indicate mean and SD from triplicates. **B** Bivariate flowcytometry analysis in MCF7 and MiaPaCa-2 cell lines following the treatment with different concentrations of INX-315 for 48 h. **C** Western blotting on the indicated proteins from MCF7, MiaPaCa-2 and 3226 cell lines following the treatment with INX-315 for 48 h. **D** Biochemical analysis to determine the differential effect of INX-315 from MB157-WT and MB157-RB-del cell lines. **E** Immunoprecipitation of CDK2 from MCF7 and MIA PaCa-2 cells following the treatment with INX-315 (500 nM) for 48 h. Coimmunoprecipitated proteins including cyclin E1, cyclin A and cyclin B1 from whole cell lysate (Input), CDK2-complex and flow-through were determined by western blotting. **F** Column graphs depicting the fraction of the indicated cyclins in complex with CDK2 and CDK1 from MCF7 and MiaPaCa-2 cell lines in the absence and presence of INX-315 (500 nM), which was determined based on densitometry analysis from the western blot in Fig. 5E. Error bars were determined based on mean and SEM. Statistical Analysis was determined based on two-tailed student t-test. In MCF7 the experimental replicates are as follows: for cyclin E1/CDK2: $n = 3$, cyclin A/CDK2: $n = 4$; cyclin B1/CDK2: $n = 3$; cyclin A/CDK1: $n = 2$, cyclin B1/CDK1: $n = 3$. The $p$ value for cyclin E1/CDK2 complex is 0.0367 and the $p$ value for cyclin A/CDK2 complex is 0.0082. For MiaPaCa-2 cells, the experimental replicates include, cyclin E1/CDK2: $n = 3$; cyclin A/CDK2: $n = 6$; cyclin B1/CDK2: $n = 3$; cyclin A/CDK1: $n = 2$; cyclin B/CDK1: $n = 3$. The $p$ value for cyclin E1/CDK2 complex is 0.0435 and the $p$ value for cyclin A/CDK2 complex is 0.0293. **G** In vitro kinase reaction to determine the catalytic activities of CDK2 and CDK1 in the presence of INX-315 at different concentrations. The kinase reactions were carried out in the presence of ATP (500 μM). The C-terminal peptide from RB was used as substrate for the kinases. The phosphorylation status of RB was determined by western blotting. **H** Densitometry analysis of pRB western blot from the Fig. 5G represents catalytic activity of CDK2 and CDK1 and plotted against the concentration of INX-315. Error bars represent mean and SEM from 3 independent biological replicates. Statistical analysis was performed based on 2-way Anova (*** represents $p$ value < 0.0001). Western blotting was done at two independent times (**C & D**). Source data are provided as a source data file.

interrogated whether this could be translated in vivo. In MCF7-xenografts bearing mice, INX-315 significantly inhibited tumor growth at a well-tolerated dose (200 mg/Kg daily) with no significant change in body weight observed (Fig. 9A–C). Histological analysis revealed that INX-315 treatment altered tumor morphology resulting in large-sized cells compared with vehicle (Fig. 9D). To further interrogate the in vivo efficacy of the combination treatment involving palbociclib and INX-315, we employed patient-derived PDAC PDX models, 1222 and 3226 and MiaPaCa-2 xenografts[52]. INX-315 elicited a prominent effect in inhibiting the growth of 3226 PDX, a modest effect in the MIA PaCa-2 xenografts, and no activity in the 1222 PDX model (Fig. 9E). Despite the differential sensitivity to single agent INX-315, combination with palbociclib enhanced the efficacy of either single-agent treatment in all three models (Fig. 9E and Supp Fig. S12A, B). Histological analysis of the tissues revealed that tumor architecture was minimally impacted following single agent treatment with palbociclib and INX-315, but the combination treatment appeared to result in fewer large-sized tumor cells (Fig. 9F, Supp Fig. S12C). Combination of palbociclib and INX-315 yielded a cooperative inhibition of RB phosphorylation in the tumor tissues derived from both the 1222 and 3226 PDX models as determined by immunohistochemical staining (Fig. 9F). As a complementary approach, multispectral immunofluorescence staining was carried out on tumor tissues derived from the 1222 PDX model, which revealed that the combination treatment inhibited RB phosphorylation in a large fraction of tumor cells as compared to the vehicle treated tumor (Fig. 9G). Furthermore, the expressions of MCM2 and KI67, which are markers of proliferation and regulated via RB, were also significantly downregulated following combination treatment (Fig. 9H). In conclusion, the combination of INX-315 with CDK4/6 inhibitors presents a promising strategy to extend the therapeutic applicability of INX-315 beyond CDK2-addicted tumor types.

## Discussion

Targeting cell cycle control with CDK-inhibitors has been the subject of extensive study since it was realized the CDK/Cyclins drive proliferation and are deregulated in cancer[53,54]. While pan-CDK inhibitors were developed many years ago, the limited therapeutic window and significant toxicity limited their clinical utility[24,55]. In contrast, CDK4/6 inhibitors have shown clinical benefits and are widely used in the treatment of HR + /HER2- breast cancer[8]. Despite the success of these drugs, disease progression occurs which is often associated with CDK2 activity[16]. Therefore, CDK2 has emerged as an ideal target for therapeutic intervention and the development of selective inhibitors are in the earliest stages of clinical development directed at *CCNE1* amplified tumors and tumors that have progressed on CDK4/6 inhibitors[32]. The data here suggests a refinement of these strategies and the need to understand the differing mechanisms of drug action which is not observed with CDK4/6 inhibitors.

DepMap dataset revealed that there is a relatively small subset of cancer cell lines that are vulnerable to loss of CDK2 and its catalytic partner cyclin E1. In canonical cell cycle machinery, cyclin D1/CDK4 initiates RB phosphorylation and activates CDK2, as observed in ER+ breast cancer models[56,57]. However, in CDK2-addicted models, RB phosphorylation is exclusively driven by cyclin E1/CDK2 activity. The CDK2 selective catalytic inhibitor INX-315 induced a potent growth arrest in the CDK2-addicted models, which closely mirrors the effect of palbociclib in CDK4-addicted ER+ breast cancer models. In each of these contexts the drugs act through RB to suppress the expression of E2F-target cell cycle genes. Interestingly, in the CDK2-addicted models INX-315 yields a dose dependent shift from G1/S to 4N cell cycle arrest, which is consistent with the prior study[32]. This difference appears to correlate with the differential binding affinity of INX-315 for cyclin E1 vs. cyclin A associated complexes[32]. Further biochemical assays confirm that in the CDK2 addicted models INX-315 exclusively induces a G1-like cell cycle arrest across all the concentrations, albeit with different DNA contents and this phenomenon is similar to *CCNA2* depletion. The exceptional response to CDK2 inhibitor in *CCNE1*-amplified models is driven by overexpression of P16INK4A, an endogenous CDK4/6 inhibitor and depletion of CDKN2A was sufficient to reverse the response to INX-315[16]. The expression of P16INK4A generates a "synthetic" dependence on CDK2, as the compensatory CDK4/6 activity for G1/S progression is already inhibited[58]. In this study by utilizing multiplexed protein staining and gene expression analysis from clinical datasets we show that a significant number of breast cancer patients harbor high levels of P16INK4A protein in conjunction with cyclin E1. The co-expression of P16INK4A and cyclin E1 is associated with poor therapeutic benefit from CDK4/6 inhibitors with very short progression-free survival[39]. Therefore, these tumors would be expected to be the most sensitive to CDK2 inhibitors as demonstrated by the in vivo sensitivity of MB157 model and should be prioritized in future clinical studies.

In contrast to the exceptional response, most tumor models demonstrate limited dependence on the CDK2 gene. In these models, catalytic inhibitors such as INX-315 and PF-07104091 inhibit cell proliferation at a modestly higher dose range. At such concentrations the CDK2 inhibition does not yield a G1-arrest, but rather leads to accumulation of cells with 4N DNA content[32]. Further analysis demonstrates that the 4N cell population indicates G2/M arrest based on the increase in CDK1 phosphorylation (Y15) and accumulation of cyclin B1. While CDK2 inhibitors were reported to possess off-target activity against CDK1, our investigation comparing the catalytic activities of CDK2 and CDK1 indicates that INX-315 selectively inhibits CDK2 at concentrations that induce cytostatic effect[59]. Additionally, comparing the cellular responses between INX-315 and a CDK1-seletive inhibitor, RO-3306 further supports that the intracellular effects of INX-315 are independent of CDK1.

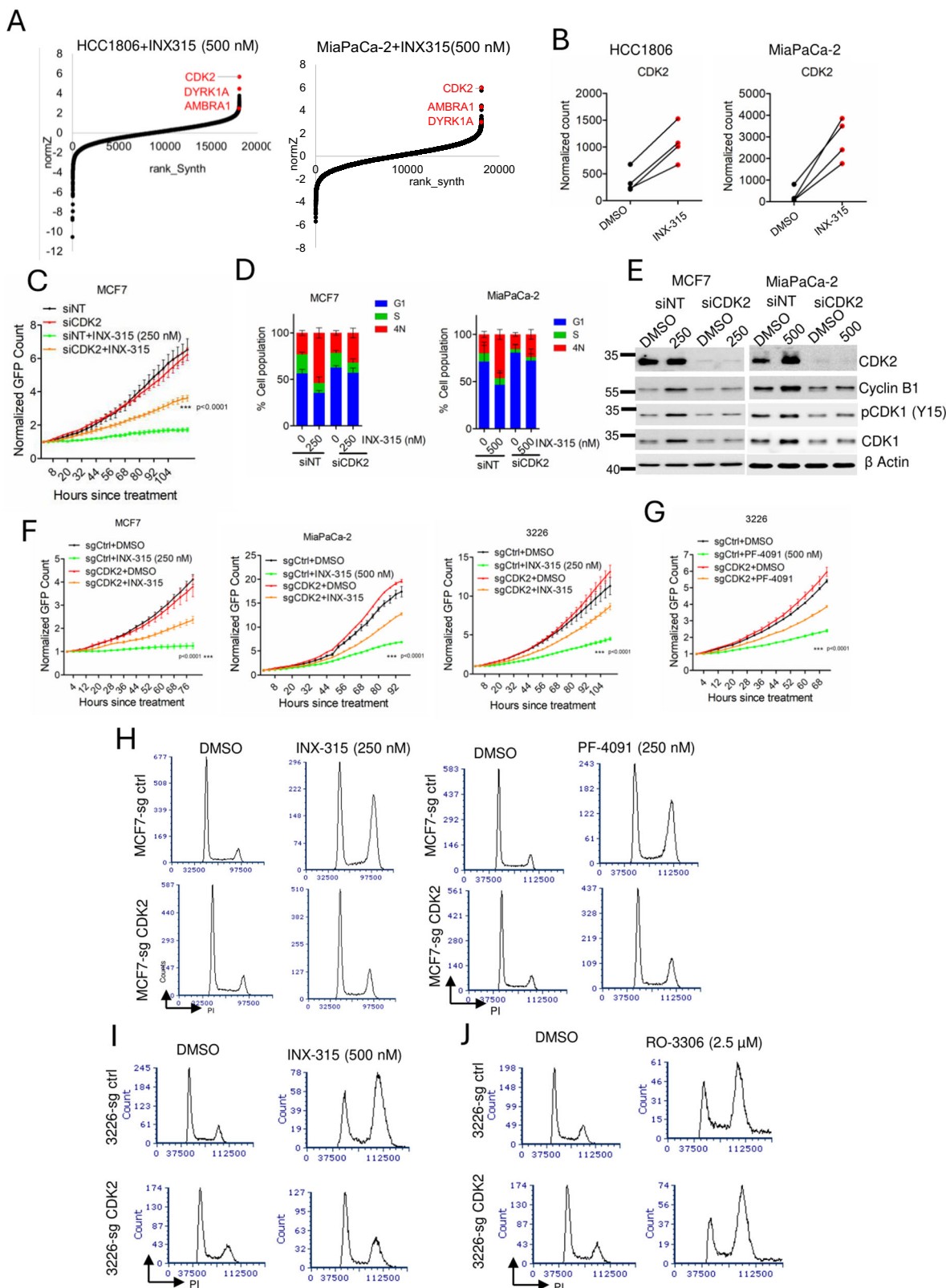

Unbiased whole-genome CRISPR screens identified the depletion of *CDK2* gene as the top driver of resistance to INX-315, suggesting that the drug-CDK2 complex is crucial for its cytostatic response and 4N block, a phenotype distinct from the gene depletion. Importantly, the deletion of CDK2 reverses the molecular features of the response to the catalytic inhibitors. The

phenomenon that kinase inhibitors can act beyond the simple gene deletion and loss of cellular target rendering drugs ineffective is not new and can be ascribed to multiple potential mechanisms including sequestering co-factors and blocking substrate accessibility[60–63]. Importantly, these data do suggest that the bulk of the response to CDK2 inhibitors is in fact mediated by

**Fig. 6 | CDK2 loss drives resistance to pharmacological inhibitors. A** DrugZ analysis from HCC1806 and MIA PaCa-2 cells identified CDK2 was positively selected following INX-315 treatment. **B** Impact of INX-315 on the normalized counts of individual guides that target CDK2 from HCC1806 and MiaPaCa-2 cell lines. **C** Live cell imaging to monitor the proliferation of MCF7 cells following the depletion of *CDK2* in the absence and presence of INX-315 (500 nM). Statistical significance was compared between siNT+INX-315 and siCDK2+INX-315. **D** Cell cycle analysis in MIA PaCa-2 cells following *CDK2* depletion in the absence and presence of INX-315 up to 48 h. Error bar represents mean and SEM from $n = 3$ independent experiments. **E** Biochemical analysis from MCF7 and MiaPaCa-2 cells on the indicated proteins following CDK2 knockdown in the absence and presence of INX-315. Western blotting was done at two independent times. **F** Differential effect of INX-

315 in MCF7-sgCtrl, MCF7-sgCDK2, MiaPaCa-2-sgCtrl, MiaPaCa-2-sgCDK2, 3226-sgCtrl and 3226-sgCDK2 cells following the treatment with INX-315. Statistical significance was compared between siNT+INX-315 and siCDK2+INX-315 sgCtrl+INX-315 and sgCDK2+INX-315. **G** Differential effect of PF-07104091 in 3226 sgCtrl and 3226 sgCDK2 cell lines based on live cell imaging. Statistical significance was compared between siNT+INX-315 and siCDK2+INX-315, sgCtrl+PF-07104091 and sgCDK2+PF-07104091. **H** Cell cycle analysis in MCF7 sg Ctrl and MCF7 sgCDK2 following the treatment with INX-315 and PF-07104091. **I** Effect of INX-315 on the cell cycle profiles of 3226 sg Ctrl and 3226 sg CDK2. **J** PI profile from 3226 sg Ctrl and 3226 sg CDK2 cells following the treatment with RO-3306 for 48 h. Experiments were done three independent times. *** represents *p* value < 0.0001 as determined by 2-way ANOVA for (**C, F & G**). Source data are provided as a source data file.

CDK2 and operable in essentially all models tested underscoring the therapeutic potential of CDK2 inhibition beyond G1/S.

To build on the general response to CDK2 inhibition, we utilized CRISPR and drug screen approaches. The CRISPR screens identified genes that modulate G2/M progression (FOXM1, CCNB1, CDK1) could potently cooperate with CDK2 inhibition, which were further validated using gene specific siRNAs in multiple models. Based on drug screens, the combination treatment involving palbociclib and INX-315 (Palbo/INX-315) resulted in synergistic inhibition of cell proliferation and in vivo tumor growth in multiple tumor models. The mechanism through which Palbo/INX-315 yields growth arrest is not mediated via a single-phase arrest but rather involves cells trapped in all phases of the cell cycle. This holistic cell cycle arrest is consistent with growing literature that CDK4/6 inhibitors are actionable beyond G1 and can serve to reinforce control over DNA replication and G2/M[64]. Our drugs screens also identified AURK and pan-CDK inhibitors that perturb G2/M transition to cooperate with INX-315. Therefore, we propose that the therapeutic application of CDK2 inhibitors could be expanded beyond the combination treatment involving CDK4/6 inhibitors.

In conclusion our study provides mechanistic evidence to illustrate the pharmacology of INX-315, which possesses dual function on cell cycle machinery to inhibit cell proliferation in both CDK2 dependent and independent models. Additionally, our study suggests numerous mechanisms to expand the utility of CDK2 inhibition as a therapeutic strategy in combination.

## Methods

We acknowledge that our research complies with all relevant ethical regulations and approved by the Institutional Biosafety Committee (IBC) at the Roswell Park Cancer Center.

### Cell culture and reagents

MCF7 (ATCC, HTB-22), CAMA-1 (ATCC, HTB-21), MIA PaCa-2 (ATCC, CRM-CRL-1420) and MB157 (ATCC, HTB-24) cells are grown in DMEM medium supplemented with 10% FBS. KURAMOCHI cells were kindly provided by Dr. Karen McLean from Roswell Park Cancer Center and grown in RPMI medium, supplemented with 10% FBS. HCC1806 (ATCC, CRL-2335) was cultured in RPMI medium supplemented with 10% FBS. Patient derived pancreatic cell lines, 1222 and 3226 were grown in Keratinocyte-free medium, supplemented with 2% FBS, EGF (0.2 ng/mL) and bovine pituitary extract (30 μg/mL) (Life Technologies, Carlsbad, CA) as described in previous study[48]. All the cell lines were maintained at 37 °C with 5% CO$_2$. The established cell lines were routinely checked to be Mycoplasma free and authenticated by STR profiling.

INX-315 was provided by Incyclix Bio (Durham, NC). The drug was reconstituted in 100% DMSO at a stock concentration of 10 mM.

### Cell proliferation assay

The proliferation of different cancer cell lines was monitored by live-cell imaging using IncuCyte and CellCyteX instruments. The cells were

transduced to stably express H2B-GFP by infecting with lenti-viral particles containing the vector, pLenti0.3UbCGWH2BC1-PatGFP in the presence of polybrene (4 μg/ml)[48]. GFP positive cells were selected by using Aria II cell sorter. Based on the GFP count, over a period of time the rate of cell proliferation was determined. The proliferation curves were generated using GraphPad Prism.

### Generation of isogenic stable cell lines

To ectopically overexpress CDK4, cells were infected with lenti-viral particles containing the vector, pLX304 that carries the open reading frame (ORF) sequence for *CDK4*. CAS9-mediated gene deletion was carried out using the vector, pL-CRISPR-EFS-tRFP that carries the guide sequences, 5′-GGTTCTTTGAGCAACATGGG-3′ and 5′-GCATGGGTGTAAGTACGAACA-3′ to target *RB1* and *CDK2*. The positive clones were selected by sorting the RFP-labeled cells using Aria II cell sorter.

### PIP FUCCI

MB157 cells were stably transfected with PIP-FUCCI using the pLenti-CMV-Blast-PIP-FUCCI vector, which was kindly provided by the Dean Tang's lab at Roswell Park Cancer Center. The cells that express both mVenus and mCherry were selected using Flow sorting. The transition between the cell cycle phases based on the fluorescent signals were monitored in real time using the IncuCyte S3 system. The resulting fluorescent signals were plotted against time using GraphPad Prism.

### Knockdown experiments

The cell lines were reverse transfected using gene specific siRNAs that include *CDK2* (Cat# L-003236-00-0005), *FOXM1* (Cat# L-009762-00-0005), *CCNB1* (Cat# L-003206-00-0005), *CDK4* (Cat# L-003238-00-0005), *CCNE1* (Cat# L-003213-00-0005), *CCNA2* (Cat# L-003205-00-0005) and *CDK1* (Cat # L-003224-00-0005). The transfections were carried out using RNAimax according to the manufacturer's protocol[65]. The final siRNA concentration used to yield efficient depletion was 12.5 nM. A non-targeting scrambled siRNA was used as a control. All the siRNAs were purchased from Horizon Discovery.

### Immunoblotting

Cells were lysed and whole cell extracts were prepared using RIPA lysis buffer (10 mM Tris HCl, pH 8.0, 1 mM EDTA, 150 mM NaCl, 1% Triton-X-100, 0.1% sodium deoxycholate, 0.1% SDS) in the presence of Halt Protease inhibitor cocktail and 1 mM PMSF[16]. The proteins were resolved on a 12% SDS-PAGE gel and transferred to nitrocellulose membrane. The primary antibodies that were utilized for probing target proteins include, pRB (S807/811) (D20B12) (Cat # 8516S), cyclin B1 (D5C10) (Cat # 12231S), CDK4 (D9G3E) (Cat # 12790S), CDK2 (78B2) (Cat # 2546S), cyclin E1 (HE12) (Cat # 4129S), pCDK1 (Y15) (9111 L), CDK1 (Cat # 77055), P16INK4A (D7C1M) (Cat# 80772) and FOXM1 (D12D5) (Cat # 5436S) were purchased from Cell Signaling Technology. The antibodies were diluted by 1000-fold in the antibody diluent solution (5% BSA in

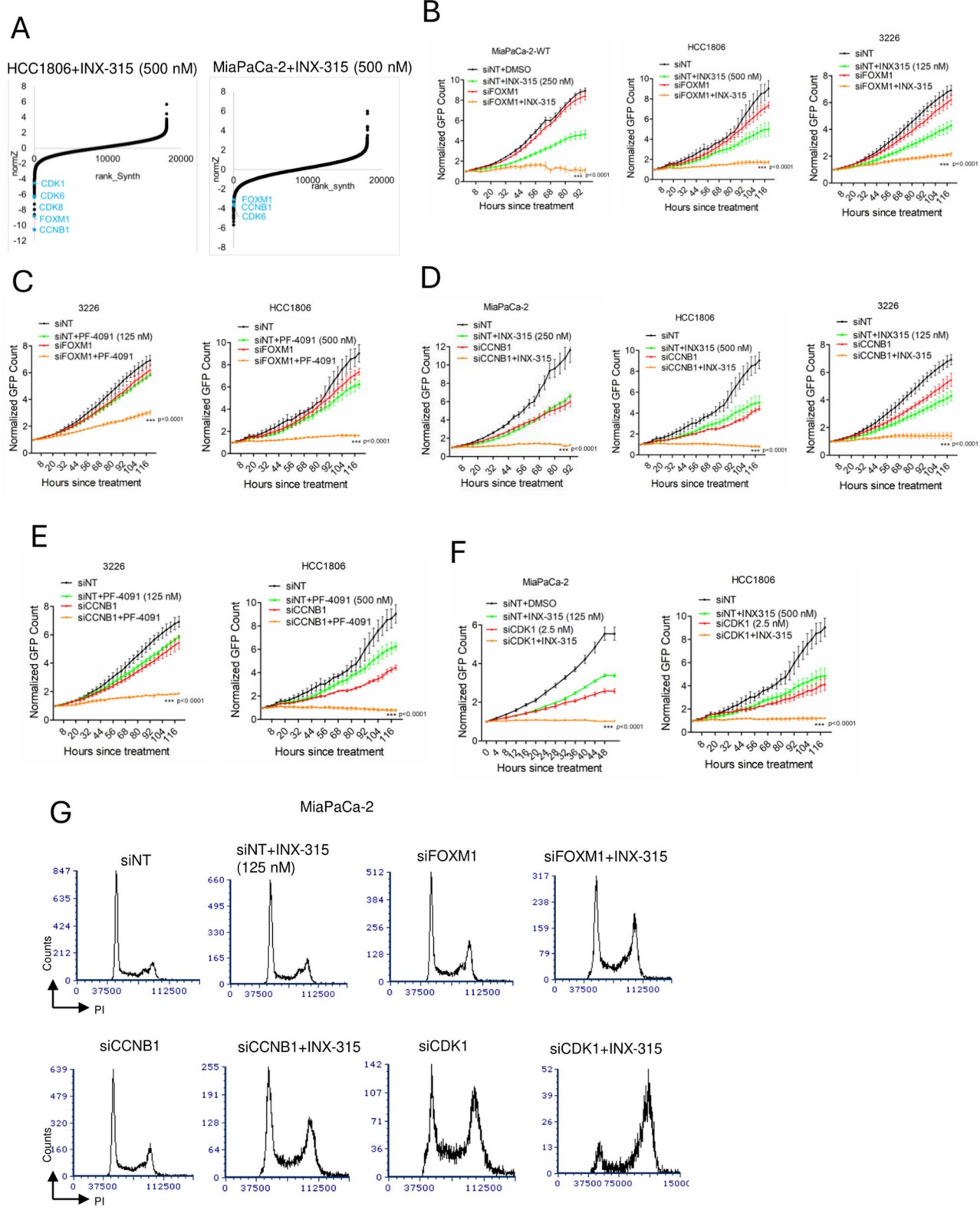

25 mM Tris HCl, pH 7.5, 150 mM NaCl, 0.1% Tween 20). Antibodies against cyclin A (Cat # AF5999) and β Actin (Cat #MAB8929) were purchased from R&D Systems (Minneapolis, MN). To validate that the antibody recognizes the specific protein, we used gene specific RNAi to delete the target protein. Western blot analysis indicates a reduction in the signal corresponding to the expected molecular weight of the protein, confirming that the antibody recognizes the specific protein

## Immunoprecipitation
Whole cell extract was prepared by using the IP-Lysis buffer (20 mM Tris-HCl, pH 8.0, 2 mM EDTA, 137 mM NaCl, 1% NP-40) in the presence

**Fig. 7 | Determinants of response to CDK2 inhibitors. A** DrugZ analysis from HCC1806 and MIA PaCa-2 cells to indicate the negatively selected genes following the selection with INX-315. **B** Live cell imaging to monitor the proliferation of HCC1806, MIA PaCa-2 and 3226 cells following RNAi-mediated *FOXM1* depletion in the absence and presence of INX-315. Statistical significance was compared between siNT+INX-315 and siFOXM1+INX-315. **C** Effect of PF-07104091 on the proliferation of 3226 and HCC1806 cell lines following the depletion of FOXM1. Error bars were determined based on mean and SD from triplicates. Statistical significance was compared between siNT+PF-07104091 and siFOXM1+PF-07104091. **D** Live cell imaging to monitor the proliferation of HCC1806, MIA PaCa-2 and 3226 cells following *CCNB1* depletion in the absence and presence of INX-315. Statistical significance was compared between siNT+INX-315 and siCCNB1+INX-315.

**E** Impact of *CCNB1* depletion on the cellular response to PF-07104091 in 3226 and HCC1806 cell lines. Statistical significance was compared between siNT+PF-07104091 and siCCNB1+PF-07104091. **F** Live cell imaging to monitor the proliferation of HCC1806 and MIA PaCa-2 cells following *CDK1* depletion in the absence and presence of INX-315. Statistical significance was compared between comparing siNT+INX-315 and siCDK1+INX-315. **G** Cell cycle analysis in MIA PaCa-2 cells to determine the impact of *FOXM1*, *CCNB1* and *CDK1* depletion in modulating the response to INX-315. All the data are presented as mean and SD from n = 3 technical replicates. Experiments were performed at n = 3 independent times. *** represents p value < 0.0001 as determined by 2-way ANOVA for **B**–**F**. Source data are provided as a source data file.

of Halt Protease inhibitor cocktail and 1 mM PMSF[16]. Antibodies against CDK2 (SC-6248) and CDK1 (SC-54) were used to pull down CDK2 and CDK1 respectively and immobilized using protein G-agarose beads (Thermo Scientific). The co-immunoprecipitated proteins were subjected to western blot analysis by probing using antibodies from Cell Signaling Technology that include cyclin B1 (Cat # 12231S), CDK2 (Cat # 2546S), cyclin E1 (Cat # 4129S) and CDK1 (Cat# 77055S). cyclin A (Cat # AF5999) was purchased from R&D Systems (Minneapolis, MN).

### In vitro kinase reactions
The catalytic activities of CDK2 and CDK1 were examined using in vitro kinase reactions[16,65]. Cells were extracted using the Kinase assay lysis buffer (50 mM HEPES-KOH pH 7.5, 1 mM EDTA, 150 mM NaCl, 1 mM DTT, 100% Glycerol and 0.1% Tween-20). Antibodies against CDK2 (SC-6248) and CDK1 (MA5-11472) from Thermo Scientific were used to pull down CDK2 and CDK1 respectively. The kinase reaction was carried out in the kinase assay buffer (40 mM Tris-HCl pH 8.0, 20 mM $MgCl_2$, 0.1 mg/ml BSA and 50 µM BSA) in the presence of 500 µM ATP along with different concentrations of INX-315. The RB C-terminal peptide was used as an exogenous substrate and the phosphorylation status was determined by western blotting[66].

### Cell cycle analysis and bivariate flow cytometry analysis
To investigate the cell cycle progression based on distinct cell cycle phases, cells were fixed using ice cold 70% ethanol overnight at -20 °C. Cells were washed with 1X PBS and incubated with RNase A (200 mg/ml) and propidium iodide (PI) (40 µg/ml). To determine the BrdU incorporation using bivariate flow cytometry analysis, cells were pulsed with BrdU for 3 h. After 3 h incubation, cells were harvested and fixed using ice cold 70% ethanol overnight at -20 °C. The fixed cells were denatured using the denaturing buffer (2 N HCl, 0.5% Triton X 100) for 30 mins at room temperature. The cells were neutralized using 0.1 M sodium tetraborate (pH 8.5) and washed with IFA buffer (1% BSA in 1X PBS). Cells were then incubated with FITC-conjugated anti-BrdU antibody for 1.5 h at room temperature. The cells were then washed with IFA buffer and resuspended in PBS in the presence of RNase A and PI. The cell cycle analysis was carried out in BD LSRFORTESSA flow cytometer and the data analysis was performed in FCS express.

### Biolayer interferometry (BLI)
Recombinant CDK2 (human, full length) and cyclin A (human Cyclin A, residues 173–465) were expressed and purified, and CDK2 was biotinylated for BLI as previously described[45]. BLI experiments were performed using an eight-channel Octet-RED96e (Santorius). CDK2 ligand was loaded onto streptavidin coated sensors at a concentration of 750 ng/mL, and sensors were dipped into 100 nM cyclin A analyte. Where indicated, 100 nM INX-315 was present in all solutions following CDK2 loading. Data were processed and fit using Octet software version 7 (Santorius) and a 1:1 binding model[45]. The first 300 s of association and 800 s of dissociation were used in the fit to determine rate constants. Four replicates of each experiment were performed.

### Mice and patient derived xenografts
NSG mice were bred and maintained at Roswell Park Comprehensive Cancer Center animal care facility. All the animal protocols including housing, tumor implantation, drug administration and animal euthanization were approved by the Roswell Park Comprehensive Cancer Center Institutional Animal Care and Use Committee (IACUC) in accordance with the NIH guide for the care and use of laboratory animals. MB157 xenografts were developed by orthotopically injecting $5 \times 10^6$ cells/mouse into the mammary fat pad of 10-week-old female NSG mice. The tumor growth was monitored and once the tumor reached 5 cm³, the tumor was excised into tiny fragments. The resulting tumor fragments were serially passaged into the mammary fat pad. Once the tumor volume reached 150 mm³, the mice were randomized in a non-blinded manner into two groups, vehicle-treated (n = 4) and INX-315 treated (n = 4) groups. INX-315 was administered via oral gavage, at a dose of 100 mg/kg once a day for 20 days. The drug was reconstituted in 100% polyethylene glycol-400 (PEG400), which was used for the vehicle treated group. MCF7 xenografts were developed in 10-week-old NSG female mice by subcutaneously injecting $5 \times 10^6$ cells/mouse that were supplemented with Estrogen pellets as described before[18]. Resulting tumors were excised and serially passaged subcutaneously. The mice bearing MCF7 xenografts were randomized into two groups: Vehicle (n = 4) and INX-315 (n = 4). Patient derived pancreatic cancer xenografts, 1222 and 3226 PDX were implanted subcutaneously into 10-week-old male NSG mice. MiaPaCa-2 xenografts were developed in 10-week-old male NSG mice by subcutaneously injecting $3 \times 10^6$ cells/mouse. The tumor growth was monitored and once it reached 150 150 mm³, the mice were randomized into 4 groups, vehicle (PEG 400), Palbociclib, which was administered via oral gavage at the dose of 100 mg/kg, INX-315 (100 mg/kg) for 20 days and the combination treatment involving palbociclib and INX-315. Mice bearing 3226 PDX were randomized as Vehicle (n = 5), Palbo (n = 4), INX-315 (n = 5) and Palbo+INX-315 (n = 6). Mice bearing 1222 PDX were randomized as Vehicle (n = 7), Palbo (n = 3), INX-315 (n = 3), Palbo+INX-315 (n = 6). Mice bearing MiaPaCa-2 xenografts were randomized as Vehicle (n = 6), Palbo (n = 4), INX-315 (n = 3), Palbo+INX-315 (n = 5). Palbociclib was reconstituted in Lactate buffer pH 4.0. The tumor growth was monitored every other day using digital calipers. The maximal tumor size permitted by our institutional review board is 20 cm³ and once the mice reach this tumor volume, the mice were euthanized. The maximum tumor burden did not exceed 20 cm³. All the treatment continued for 20 days. Any mice that died during the course of treatment was removed from the analysis.

### Histological analysis
Tumor tissues that were excised from the mice were fixed in 10% Formalin followed by processing and paraffin embedding. The embedded tissues were serially sectioned at 4–6 µm using the

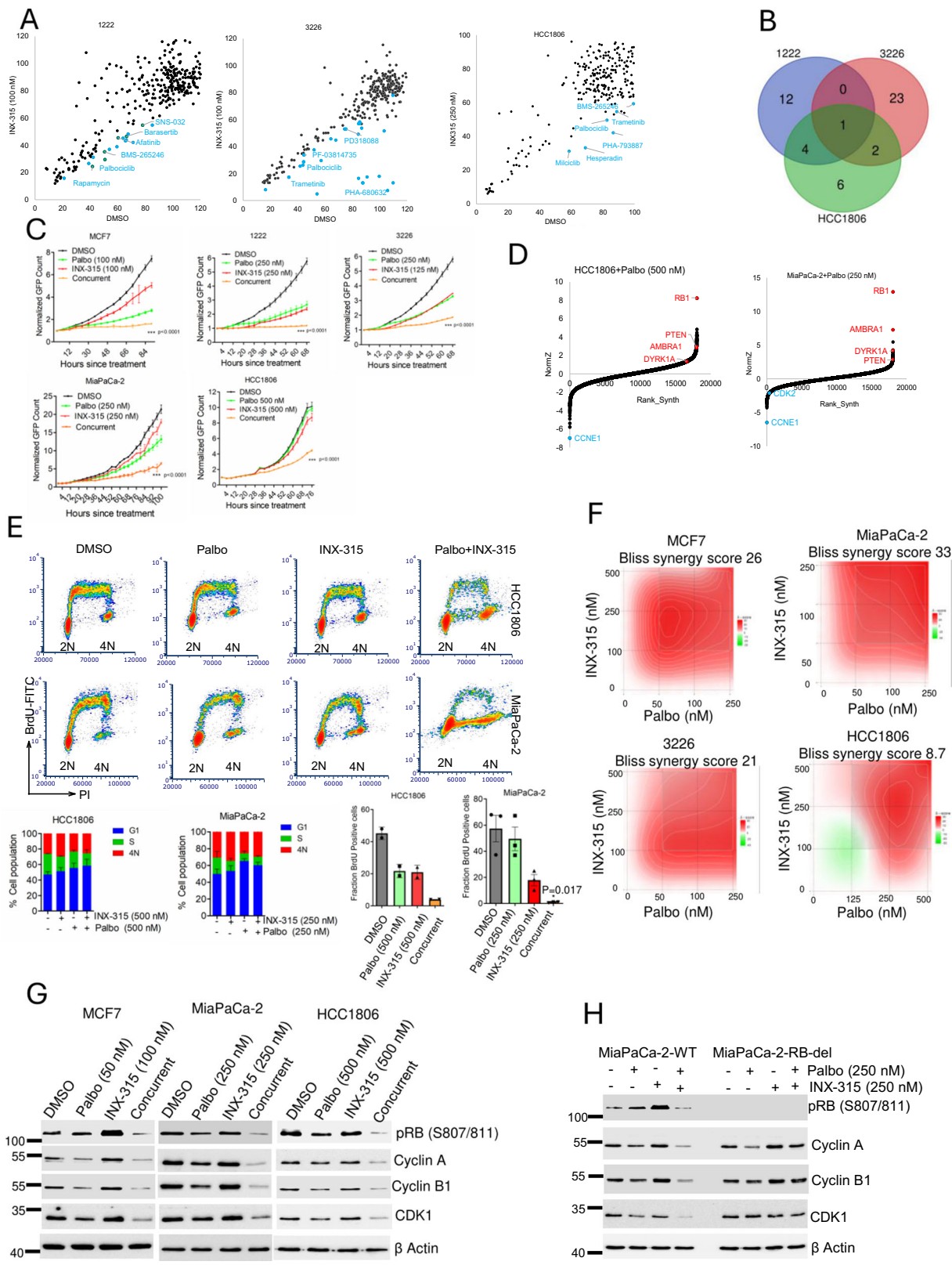

standard procedures and were subjected to Hematoxylin and Eosin (H&E) staining[48]. Immunohistochemical staining was performed using the pRB (S807/811) antibody (Cat # 8516) (Cell signaling Technologies). Slides were scanned using the Vectra Polaris Instrument.

**Multi spectral immunofluorescence staining (MIF)**

The immunofluorescent staining was carried out on formalin-fixed Paraffin-embedded (FFPE) tissue sections using the Opal 6-Plex Detection kit (AKOYA Biosciences, Cat # NEL821001KT) as described in our previous study[44]. The MIF panel consisted of the following

**Fig. 8 | Enhanced efficacy of INX-315 in combination with CDK4/6 inhibition.**
**A** Scatter plot analysis in 1222, 3226 and HCC1806 cell lines following a combinatorial drug screen. *X* axis indicates the efficacy of individual drugs from the library in combination with DMSO. *Y* axis indicates the efficacy of individual drugs in combination with INX-315. **B** Venn Diagram illustrating the total number of drugs from cell line that cooperatively inhibited cell proliferation in combination with INX-315. **C** Live cell imaging to monitor the proliferation of the indicated cell lines following the treatment with palbociclib in combination with INX-315. Error bars represent mean and SD from triplicates. Experiments were done at $n = 3$ independent times. ***$p < 0.0001$ as determined by 2-wat ANOVA, comparing the INX-315 and Palbo/INX-315 treated groups. **D** DrugZ analysis from MIA PaCa-2 and HCC1806 cells indicating the positively and negatively selected guides following the selection with palbociclib. **E** Bi-variate flow cytometry analysis on MIA PaCa-2 and HCC1806 cells to indicate the combined effect of palbociclib and INX-315 on the BrdU incorporation within the S-phase population of cells. The experiment was done at $n = 3$ biological replicates for MiaPaCa-2 cells and $n = 2$ biological replicates for HCC1806 cells. Stacked bar plots represent the population of cells at each phase of the cell cycle. Error bars were determined from mean and SD from $n = 4$ biological replicates for MiaPaCa-2 and $n = 3$ biological replicates for HCC1806 cells. Fraction of cells within S-phase population incorporating BrdU was quantified. Statistical analysis was carried out based on two-tailed student t-test comparing the INX-315-treated and concurrent -treated groups. The *p* value for MiaPaCa-2 is 0.017. **F** The synergistic interaction between Palbo and INX-315 is based on BrdU incorporation. The synergistic interaction was determined using Synergy Finder, which calculates the Bliss synergy score. **G** Biochemical analysis on the indicated proteins from MCF7, MIA PaCa-2 and HCC1806 cell lines following the treatment with palbociclib in combination with INX-315 up to 48 h. MCF7 cells were treated with palbo (50 nM) and INX-315 (100 nM). MIA PaCa-2 cells were treated with Palbo (250 nM) and INX-315 (250 nM). HCC1806 cells were treated with Palbo (500 nM) and INX-315 (500 nM). **H** Immunoblotting in MIA PaCa-2-WT and MIA PaCa-2-RB-del cell lines following the treatment with palbociclib (250 nM) in combination with INX-315 (250 nM) up to 48 h to determine the differential impact on cell cycle proteins. Western blotting was done at two independent times (**G & H**). Source data are provided as a source data file.

---

antibodies, cyclin A (Y193, abcam, 1/600, Opal 540), Cytokeratin (AE1/AE3, Agilent DAKO, 1/300, Opal 480), Ki67 (SP6, abcam, 1/200, Opal 780), pRB ser807/811 (D20B12, Cell Signaling, 1/250, Opal 690), RB (4H1, Cell Signaling, 1/150, Opal 620) and MCM2 (RBT-MCM2, BioSB, 1/750, Opal 650).

To determine the predictive biomarkers of response, mIF staining was performed on patient derived breast cancer TMAs. Following the QC, 197 patient tissues were utilized for further analysis. The biomarker panel comprised of the following antibodies, cyclin A (Y193, abcam, 1/600, Opal 540), cyclin E1 (EP435E, abcam, 1/500, Opal 520), Cytokeratin (AE1/AE3, Agilent DAKO, 1/300, Opal 480), RB (4H1, Cell Signaling, 1/150, Opal 620), Geminin (EPR14637, abcam, 1/200, Opal 690), P16INK4A (6H12, Leica Biosystems, RTU, Opal 780). Following the staining, slides were imaged on the PhenoImager HT® Automated Quantitative Pathology Imaging System (AKOYA Biosciences). Further analysis of the slides was performed using inForm® Software v2.6.0 (AKOYA Biosciences). Based on the phenotype cell data, the percent of cell expressing a given cell cycle marker was determined, which was used to calculate the correlation matrix with the correlation function from base R (v4.3.2) with method set to "pearson".

## DepMap data analysis

To cluster cancer cells depending on their vulnerability to loss of selective cell cycle genes, we used gene set (CCND1, CCND2, CCND3, CCNE1, CDK2, CDK4, CDK6) and data file (CRISPRGeneEffect.csv) downloaded from DepMap Portal (23Q4, https://depmap.org/portal/). Cell lines from all tumor types ($n = 1100$) are used for k-means clustering with the parameter "column_km" in the function "Heatmap" from the ComplexHeatmap (v2.18.0) Bioconductor package set to 6, based on elbow plot using the function "fviz_nbclust" from the factoextra (v1.0.7) r package. We identified differentially expressed genes between selected clusters using t-test with gene expression data file from DepMap Portal (OmicsExpressionProteinCodingGenesTPMLogp1.csv). The output file from t-test was used for making volcano plot using the Bioconductor package, EnhancedVolcano (v1.14.0).

## Transcriptome analysis

MB157, MCF7 and HCC1806 cell lines were treated with INX-315 at a concentration of 100 nM up to 48 h. Total RNA was extracted using the Qiagen RNeasyplus kit and the RNA quality was evaluated using the RNA6000 Nano assay and with the Agilent 2200 TapeStation (Agilent, CA, USA)[48]. To obtain nonribosomal RNA transcripts,

cDNA synthesis was performed using random hexamers[48]. Sequencing libraries were prepared using the DriverMap Human Genome-Wide Gene Expression Profiling Sample Prep Kit hDM18Kv3 (Cellecta Inc., CA, USA). Anchor PCR was performed to avoid primer dimer formation, and the PCR conditions are as follows, denaturation for 5 mins at 95 °C, followed by 15 cycles of (95 °C – 0.5 min, 68 °C – 1 min, 72 °C – 1 min) with a final extension at 72 °C for 10 mins. The PCR product was purified using SPRI (Agentcourt, 1:1 sample: reagent ratio) and the concentration was determined based on the Qubit fluorescence assay (Qubit dsDNA HS Assay Kit, ThermoFisher Scientific, MA, USA)[48]. Target-enriched RNAseq libraries were analyzed on an Illumina NextSeq 500 sequencer using a NextSeq500/550 High Output v2 Kit (75 cycles) according to the standard manufacturer's protocol (Illumina, CA, USA) and the alignment was carried out using STAR v2.7.10b, which generates gene-level read counts. Differential gene expression analysis was performed using the EdgeR software and selected based on the p values and log 2-fold change.

## CRISPR screening

The Toronto Knockout (TKO) CRISPR library version 3 was packaged into lentiviral particles and were used to infect the HCC1806, and MiaPaCa-2 cell lines[47]. Following the infection, the cells were selected using puromycin (5 μg/ml) to get a mutant pool up to 200-fold coverage[48]. The positive clones were further expanded in the absence and presence of INX-315 (500 nM) up to 5 passages. The palbociclib selection was carried out at 250 nM for MiaPaCa-2 cells and 500 nM for HCC1806 cells. Following the 5th passage, the cells were harvested and subjected to genomic DNA extraction using the Wizard Genomic DNA purification kit (Promega, A1120). For the downstream processing the genomic DNA is subjected to a 2-step PCR to enrich the guide RNA region in the genomic DNA and amplify the guide RNA using Illumina TruSeq adapters with i5 and i7 indices[48]. This PCR generates libraries that were sequenced on Illumina's NexSeq platform to get the Fastq files. The adapters are removed using Trim Galore (v0.6.7, https://github.com/FelixKrueger/TrimGalore)[48]. The resulting Fastq files were processed using the MAGeCK pipeline that gives us the count files, which is further used for the DrugZ analysis.

## Clinical dataset analysis

The metabric breast cancer gene expression data was downloaded from cBioPortal for cancer genomics website (https://www.cbioportal.org/). We selected TNBC subset with the following criteria: ER_STATUS == 'Negative', HER2_STATUS == 'Negative', PR_STATUS == 'Negative'. We then created the correlation scatter plot for CDKN2A and

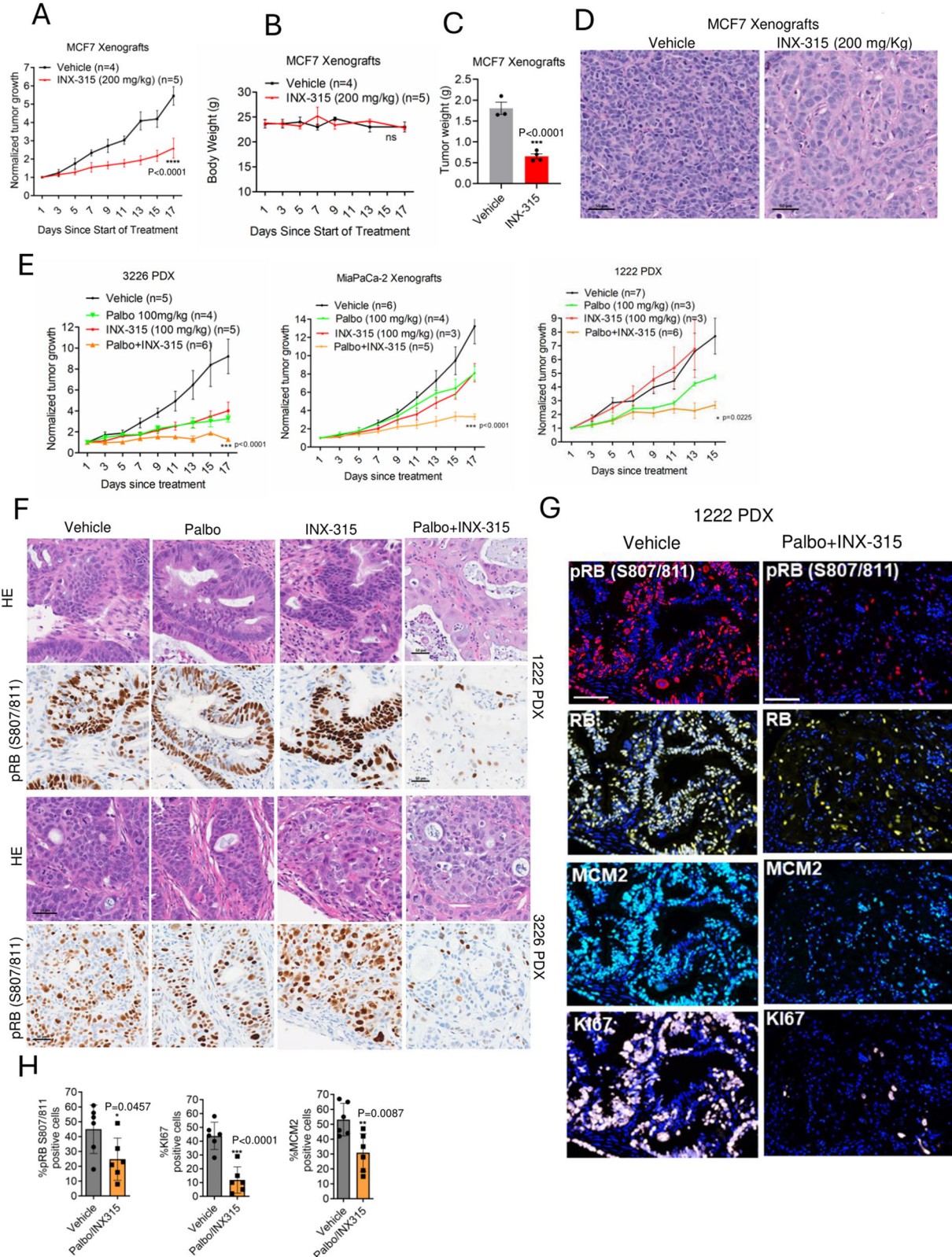

CCNE1 using the ggscatter function from ggpubr (v0.6.0) r package. PALOMA2 and PEARL raw read count data were downloaded from Gene Expression Omnibus (GEO), with accession numbers GSE133394 and GSE223700, respectively[42,43]. We normalized these raw read counts using the edgeR bioconductor package (v3.40.2). Cancer subtypes are calculated using the AIMS R package

(v1.30.0). All data analysis is performed under the R statistical environment v4.3.2.

**Reporting summary**
Further information on research design is available in the Nature Portfolio Reporting Summary linked to this article.

**Fig. 9 | In vivo anti-tumor efficacy of INX-315 in combination with palbociclib. A** Relative tumor growth on MCF7 xenografts treated with vehicle ($n = 4$) and INX-315 (200 mg/Kg) ($n = 5$). **B** Change in mice body weight that were treated with vehicle ($n = 3$) and INX-315 ($n = 5$). Error bars represent mean and SEM for (**A & B**). *** represents $p$ value < 0.0001 (**A**), $p$ value = 0.5564 (**B**) as determined by 2-way ANOVA comparing vehicle- and INX-315-treated groups. **C** Column graph depicting the tumor weights from MCF7 xenografts that were treated with vehicle ($n = 3$ mice) and INX-315 ($n = 4$ mice). Error bars represent mean and SEM. Statistical significance was determined by two-tailed unpaired student t-test. (*** $p < 0.0001$) comparing vehicle- and INX-315-treated groups. **D** Representative images of the H&E staining on the tissues from MCF7 xenografts that were treated with vehicle and INX-315. Scale bar represents 50 microns. **E** Relative tumor growth on MIA PaCa-2 xenografts and two different PDAC derived PDX, 3226 and 1222 to determine the effect of INX-315 in combination with palbociclib. The treatment groups for 3226 PDX are vehicle ($n = 5$), Palbo ($n = 4$), INX-315 ($n = 5$) and Palbo/INX-315 ($n = 6$). The treatment groups for MiaPaCa-2 xenografts are vehicle ($n = 5$), Palbo ($n = 4$), INX-315 ($n = 3$) and Palbo/INX-315 ($n = 5$). The treatment groups for 1222 PDX are vehicle ($n = 7$),

Palbo ($n = 3$), INX-315 ($n = 3$) and Palbo/INX-315 ($n = 5$). Error bars represent mean and SEM. *** represents $p$ value < 0.0001 (3226 and MiaPaCa-2), * $p$ value = 0.0225 (1222 PDX) as determined by 2-way ANOVA, comparing the INX-315 and Palbo/INX-315-treated groups. **F** Representative images of the H&E staining and immunohistochemical staining of pRB on the 1222 and 3226 PDX tissues that were treated with vehicle, palbociclib, INX-315 and Palbo/INX-315 groups. The staining was done on tissues derived from $n = 3$ mice for each condition. The scale bar represents 50 microns. **G** Representative images of MIF staining on the indicated proteins from the 1222 PDX that were treated with vehicle and INX-315 in combination with palbociclib. The scale bar represents 50 microns (**H**) Column graphs depicting the fraction of tumor cells expressing the indicated proteins from the vehicle (2 technical replicates, $n = 3$ mice) -and Palbo/INX-315-treated groups (2 technical replicates, $n = 3$ mice). Error bars were determined from mean and SEM. * represents $p$ value 0.0457 (pRB (S807/811)), *** indicates $p$ value < 0.0001 (KI67) and ** represents $p$ value 0.0087 (MCM2) as determined by two-tailed unpaired student t-test. All the comparisons were made between the vehicle- and Palbo/INX-315 treated groups. Source data are provided as a source data file.

## Data availability

The raw data generated from the transcriptome analysis and CRISPR screen for this study are deposited in GEO under accession GSE287204 and GSE287048 (https://www.ncbi.nlm.nih.gov/geo/query/acc.cgi) respectively. The processed data for both the transcriptome analysis and CRISPRs screen can be found in Supplementary Data 1–3. The METABRIC data is publicly available from cBioportal (https://www.cbioportal.org/study/summary?id=brca_metabric). The gene expression data from PALOMA2 and PEARL clinical trials are publicly available in the GEO database under accession codes GSE133394[42] and GSE223700[43]. Raw western blot images and data tables that were generated in this study are submitted in the source data file. All the related data could be also accessed via the Figshare link (https://doi.org/10.6084/m9.figshare.25770186). All remaining data can be found in the Article, Supplementary and source Data files Source data are provided with this paper.

## Code availability

No code was generated in this study.

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

## Acknowledgements

The authors thank all members of the laboratory group, colleagues and collaborators from Incyclix Bio in the discussion and preparation of the manuscript. We thank Dr. Dean Tang's laboratory at Roswell Park Cancer Center for kindly sharing the PIP-FUCCI construct and the CDK1 inhibitor, RO-3306. This work was supported by grants awarded to Erik S. Knudsen and Agnieszka Witkiewicz from the National Institutes of Health, National Cancer Institute (CA267467 and CA211878). Moreover, this work was supported by the National Cancer Institute (NCI) grant P30CA016056 involving the use of Roswell Park Comprehensive Cancer Center's Advanced Tissue imaging (ATISR), Drug Discovery Core (DDCSR), Experimental tumor models (ETM) and Genomics shared resources.

## Author contributions

Study concepts and design: V.K., C.F., A.T., P.R., S.M.R., K.M., E.S.K. and A.K.W. Acquisition of data: V.K., C.F., J.W., Y.W., A.D., H.R., A.T., J.B., J.S., P.R. and S.P. Analysis and interpretation of data: V.K., C.F., J.W., Y.W., A.P.D., H.R., S.P., A.T., P.R., E.S.K. and A.K.W. Study supervision: E.S.K. and A.K.W.

## Competing interests

The contributing authors, Alec Trub, John Bisi, Jay Strum and Patrick Roberts are employees of Incyclix Bio, Durham, NC, USA and declare competing interests. However, Incyclix Bio did not provide any financial support for this study.
