## [Peer Review file · Nature Communications]

Discrete vulnerability to pharmacological CDK2 inhibition is governed by heterogeneity of the cancer cell cycle

Corresponding Author: Dr Erik Knudsen

Version 0:

Reviewer comments:

Reviewer #1

(Remarks to the Author)

This is an interesting analysis of the dependence of CDK2 and the effect of the CDK2 inhibitor INX-315 in human cancer cells. As previously known, few cancer cells display a strong dependence on CDK2. By using a combination of RNA interference and INX, the authors propose that co-expression of p16INK4a and cyclin E may be useful as biomarkers for response to CDK2 inhibitors. Accordingly, CDK4 and CDK2 ablation or inhibition cooperate in specific models. By studying cell lines with lower response to CDK2 inhibitors, additional phenotypes such as 4N arrest are observed when using higher doses of the inhibitor. Since these phenotypes are not observed upon CDK2 genetic ablation, a hypothesis is proposed that suggests additional effects of CDK2 (more obvious when CDK4 is also inhibited) in other phases of the cell cycle.

The study is very well executed and there are many interesting observations and conclusions in the first part of the manuscript, although novelty perhaps is more reduced. Data in the second part (Mechanistic impact of CDK2 inhibition beyond G1/s) are less solid, leading to mechanistic conclusions that are mostly speculative. I would suggest to moderate most of the conclusions in this section unless the authors want to investigate more in detail these phenotypes.

Specific comments and questions:

The authors suggest senescence in MB157 cells citing ref. 38. Whereas Fig. S2A look senescent, it may be important to verify this, especially when discussing increasing DNA content, a phenotype also observed as a consequence of the DNA damage response (see below).

There is problem in the manuscript with the use of "4N" or "G2/M" arrest. These concepts are used interchangeably in the manuscript, leading to possible misinterpretations. For instance, the authors use these data to support that cells are senescent (G1 4N arrest; several sections of the text and Ref. 38) or to support the hypothesis that arrested cells are in G2/M (lines 424-425; 434-435; Fig. 5E) supporting the effect of these inhibitors in other phases of the cell cycle (lines 435-438, 490-491, abstract, etc.). It is actually quite possible that all these cells are arrested in 4N G1, but the only way to discriminate between 4N G1 and G2/M is to use additional G2 or prometaphase markers together with DNA content.

The authors claim that "INX-315 suppresses proliferation through a mechanism that involves CDK2-mediated cyclin sequestration". I would say that data in the manuscript do not support this conclusion, or that this possibility has not really been addressed in the manuscript. For instance, lines 371-374 indicate that the effect of high doses of INX is reversed by depletion of CDK2. However, this is only true for MiaPaCa (which arrested in G1), but not for MCF7 (with arrest in G2/M)(Fig. 4B,C). In agreement, whereas CDK1 complexes did not change in MiaPaCa, it seems that CDK1 is increasingly bound to Cyclin B1 in MCF7 (in the only WB-IP shown; CDK1 levels decrease whereas Cyclin B1 increase; replicates of this experiment should be carefully quantified and presented).

Together with other reasons discussed in the manuscript, it is very difficult to discriminate to what extent G2/M-4N G1 arrest depends on cyclin sequestration or partial CDK1 inhibition. In the absence of CDK2, very partial inhibition of CDK1 may have strong effects. Most of the assays in the manuscript (e.g. Fig. 4C,D,E,F, or the subsequent analysis of G2/M genes) are not useful to discriminate between the hypothesis of cyclin sequestration, favored by the authors, versus partial effects in CDK1. The fact that INX-315 induces G1 arrest at low doses and 4n arrest at higher doses "in the same cell line" (e.g. Fig. 2A), actually favors the option of inhibition of additional kinases at higher doses. In the same cell line, CDK2 inhibition and cyclin sequestration should still arrest first in G1, whereas 4n arrest could easily be explained by CDK1 inhibition. Moreover, the combination with Aurora/FoxM1 etc. inhibition does not support the hypothesis of G2 functions as these treatments also cooperate with palbociclib in multiple tumor scenarios. To analyze this in most detail, Fig. 4F should be performed with CDK1-Cyclin A2 and CDK1-Cyclin B1 in parallel to CDK2.

CDK1 depletion could also be tested although it is understandable that these assays depleting CDK1 may be difficult due to the essentiality of this gene (that may also explain why does not show up as a hit in Fig. 4B. It would be interesting to know

the position of CDK1 and other CDKs or CDK-like kinases in that plot). Can the authors provide the results of the CRISPR screening in Fig. 4B (ranking of genes/sgRNAs) as Supplementary Information?

In general, it is not easy to understand why the authors think that higher doses of INX have different phenotypes by uniquely affecting the very same kinase. The IC50 for CDK2-CycE/A is in the range of 0.6-2.5 nM and the IC50 for CDK1-CycB is about 30 nM (10-fold; about the same difference than cells sensitive to 30 nM vs. 500 nM). In addition, I did not find data for the IC50 on CDK1-CycA but it will very likely be more similar to CDK2-CycA. As discussed above, demonstrating that these effects at high doses do not depend on CDK1 inhibition would require direct evidence rather than indirect correlations.

Lines 344-346. It may be informative to indicate the percentage of tumors with these two markers.

Why using cyclin A1 (meiotic cyclin) for biochemical studies whereas cyclin A2 is targeted in the genetic assays?

Is it a bit suppressing that cyclin A is not bound to CDK1 in Fig. 4E. Does this antibody recognize CCNA1, CCNA2 or both?

Very difficult to see labels in some critical plots (Fig. 2E,G, 3A, 4B, 5A, 5D, etc)

References 21 and 23 are duplicated.

Some references are incomplete (8, 18, 31, 41)

Reviewer #2

(Remarks to the Author)

This study focuses on the outcomes of CDK2 inhibition in human cancer cells. The subject of the study is very timely, as CDK2 inhibitors are now used in several clinical trials. The authors identified "CDK2-addicted cell lines" (exceptional responders) in which CDK2 inhibition arrested cell growth and cell lines relatively insensitive to CDK2 inhibition (called by authors "cluster 2" cells). The main take-home messages of the study are: (1) "CDK2-addicted cell lines" have high p16INK4A expression. This indicates that p16 expression can be used as a predictor of a strong response to CDK2 inhibitors; (2) Using CDK2 inhibitor INX-315, they performed a screen for knockouts that make cells resistant to CDK2 inhibition and identified CDK2 as a top hit; (3) In a screen for knockouts that sensitize cells to CDK2 inhibition they identified G2-specific genes, such as FOXM1 or cyclin B. (4) They demonstrate a strong synergistic effects of CDK2 and CDK4 inhibition (Palbociclib) in cancer cell lines and patient-derived xenografts .

Observation #1 is very novel, interesting and of important clinical value. Also observation #3 is interesting and unexpected. Regarding #2, I found it very confusing that knockout of CDK2 (which should phenocopy CDK2 inhibition) actually makes cells resistant to CDK2 inhibition. The authors show that INX-315 stabilizes cyclin-CDK2 complexes. Does it mean that CDK2 acts as an inhibitor of cyclins?. The authors propose that CDK2 inhibitor may "trap" cyclin A in an inactive complex, whereas CDK2 knockout might allow formation of cyclin A-CDK1. They may be true, but needs to be demonstrated. Observation #4 is very solid but not unexpected - inhibition of CDK4/6 plus CDK2 is expected to block division of cancer cells.

Specific points:

1. Fig. 2A. INX-315 at higher concentrations arrested "CDK2-addicted" cell lines in G2/M. Could this be due to off-target effects, for instance on CDK1? This needs to be addressed.
2. Similarly, "cluster 2" cell lines eventually arrest cell proliferation if sufficiently high concentration of CDK2 inhibitors are used – could this be an off target effect?
3. Along the same lines: Knockout of RB1 in "CDK2-addicted" cell lines makes them insensitive to low concentration of CDK2 inhibitor, but these cells still respond to higher concentrations (Fig. 2F). Could this be off-target effect?
4. Fig. 1D. Inhibition of CDK2 inhibited expression of cyclins A and B1. This is likely consequence of cell cycle arrest – this should be clearly stated.
5. In the part about high p16 levels conferring sensitivity to CDK2 inhibition. The authors overexpressed p16 and show that this makes cells sensitive to CDK2 inhibition. But p16 overexpression already had a strong effect on cell growth (Fig S3C). It would be beneficial to evaluate how the genetic depletion of CDKN2A will affect response to CDK2 inhibition. I would suggest employing a CRISPR-Cas9 mediated knockout of CDKN2A in 2-3 models belonging to 'cluster 3' to evaluate this hypothesis.
6. In Fig.5G authors have shown how Palbociclib and INX-315 treatment changes the protein level of cell cycle components. It is surprising that treatment with Palbociclib in a CDK4-dependent model - MCF7 - does not lead to downregulation of RB1 phosphorylation in the presented data.
7. Most of experiments were done with a CDK2 inhibitor INX-315. It would be good to confirm at least some key results with other CDK2 inhibitors.

Reviewer #3

(Remarks to the Author)

Kumarasamy et al describe vulnerabilities and biomarkers of CDK2 inhibition using cancer cell line models. An exceptional sensitivity to CDK2 inhibition (INX-315) is conferred through a combination of RB-dependent transcriptional responses and the expression of P16INK4A. In other cell lines, the CDK2 inhibitor INX-315 suppresses proliferation through a mechanism that involves CDK2-mediated cyclin sequestration. Cooperation of CDK2 plus CDK4/6 inhibition is also studied.

The study is well performed and rises some questions that would be worth addressing.

Figure 1A is a graphical representation of a DEPMAP analysis showing vulnerabilities to cell cycle kinases (CDK2/4/6) and cyclins (D1/2/3 & E1). Would it be interesting to show a lane including cyclin A, since it also complexes with CDK2? Is there an overlap of cell lines with CDK2 / cyclin A lethality or is it a different cluster to the one of CDK2 / cyclin E1?

Also in this Figure 1A, it would be interesting to indicate the type of cancer cell line right below the cluster marks: breast (ER+, TNBC, HER2+), ovarian, gastric, etc... to capture any disease-enrichment within the clusters; e.g. is cluster 2 enriched in ER+ BC and cluster 3 enriched in ovarian cancer and TNBC cell lines?

The volcano plots barely readable.

Figure 3A shows a cluster comparison btw cluster 3 vs cluster 2 where CDKN2A and CCNE1 appear to be the enriched factors btw CDK2/cyclin E1- vs CDK4/cyclin D1-dependent clusters. Does the comparison btw cluster 3 vs cluster 6 (CDK6/cyclin D1-dependent), also render CDKN2A as differential factor?

Figures 3C/D show how overexpression of p16 renders HCC1806 cells to be dependent on CDK2 for pRb phosphorylation. Does p16 overexpression also sensitize HCC1806 cells to CDK2i?

Figure 3G shows two examples of TNBC tumors with high p16 and cyclin E1. Figure 3H shows an analysis of coexpression of biomarkers. How many tumors out of 139 TNBCs were high p16 and cyclin E1 protein expression that are RB positive. Any chance to show data from ER+ BC tumours? Several clinical trials have reported expression or protein data: e.g. PALOMA2/3 (PMID: 35974168) and PEARL (PMID: 33385521) have run the the HTG Edge-Seq Oncology Biomarker Panel; the Preoperative trials (POP, PMID: 29893769 and Abema-POP; PMID: 36071033) from Institute Gustave Roussy have run IHC of these biomarkers in metastatic/primary ER+BC.

Figure 4A shows sensitivity of the ER+ BC cell line MCF7 to high-dose CDK2i. Is the high-dose CDK2i tolerable for in vivo use in mice? Please provide dose response of MCF7 xenografts.

Figure 4A. Please for consistency, provide experiment for HCC1806

Figure 4B. What are the other hits positively selected following INX-315 treatment? Any specific cyclin? Is RB1 found? Please provide list as supplementary table(s).

Figure 4E. Please provide evidence that CDK2 kinase activity is reduced upon treatment with INX-315.

Figure 5E. Please reason why are there different concentrations of INX-315 and palbo used. Please show consistent cell cycle color coding across the manuscript

Figure 6D. please cite in the text; the figure is not mentioned

In the discussion (approx. line 480), it would be interesting to reason that p16 has been identified as mechanism of resistance to CDK4/6i in ER+ BC, so p16 can help identify patients with ER+ tumors that might be sensitive to target with CDK2i (PMID: 36071033)

The discussion states (line 501): "The CRISPR screens identified genes that regulate CDK1 (e.g. CCNB1) and coordinate G2/M progression (FOXM1) that potentially cooperate with CDK2 inhibition. Presumably, drug screens identified aurora kinase inhibitors that perturb G2/M progression to cooperate with the CDK2 inhibitor, INX-315." Were any chemotherapeutic agents present in the drug screens and provided rationale for combination approaches that are useful for the clinical development of CDK2i?

Version 1:

Reviewer comments:

Reviewer #1

(Remarks to the Author)

The authors have made a significant effort to address all reviewers' comments and the manuscript is much more solid now. It describes very interesting data using solid and elegant assays, provides nice discussion and data may have clinical potential. I still believe that the three evidences provided to indicate the selectivity of INX-315 (1) more potent on CDK2 than CDK1; 2) distinct responses when compared to CDK1i; c) cooperation with CDK1, CCNB1 or FOXM1 ablation) do not exclude the possibility that INX315, at high doses, induces 4N arrest by ""partially"" inhibiting CDK1 as well as CDK2 kinase activity in a direct manner. Yet, I understand the difficulties of discriminating between direct and indirect partial inhibition of CDK1.

Reviewer #2

(Remarks to the Author)

My concerns have been satisfactorily addressed.

Reviewer #3

(Remarks to the Author)

The authors have addressed the reviewer questions

REBUTTAL: We appreciate the reviewers for their constructive feedback on our manuscript and their overall enthusiasm for the topic. As noted below, we have addressed all the issues raised by revising the text, rephrasing the conclusions, and incorporating important new data. Below is a point-by-point discussion of the revisions to address criticisms and we have noted the additional data present in the revised manuscript. The reviewer's text is copied verbatim, and our response is provided in blue font.

Since the reviews are relatively long, we provide a brief summary of key data that addresses the three major concerns raised by the reviewers. We have included multiple new data in the manuscript as two additional figures and supplementary information. We also discuss high-level text alterations.

Summary

Mechanism of action of CDK2 inhibitors: The data in the manuscript indicates two distinct, context dependent cellular responses to CDK2 inhibition. (i) In CDK2- addicted models, pharmacological inhibition exclusively induces a G1-like arrest, which is demonstrated by multiple experimental approaches; (ii) In CDK2- independent models, the CDK2 inhibitors drive cell cycle arrest beyond the G1-phase. We have rigorously expanded our investigation into these CDK2-independent models, uncovering new insights into the pharmacology of CDK2 inhibitors as follows:

1. In models that do not require CDK2 for proliferation, pharmacological inhibition of CDK2 suppressed cell proliferation by inducing cell cycle arrest with a 4N DNA content. This state harbors high levels of RB phosphorylation, Cyclin A and B1, and tyrosine phosphorylation of CDK1 (Fig. 5C).
2. Unbiased Genome-wide CRISPR screens revealed an unexpected finding that CDK2 loss is a key mediator of resistance to INX-315 in such models (Fig. 6A & 6B).
3. Depletion of CDK2 reverses the drug-induced G2/M block and CDK1 tyrosine phosphorylation, enabling the recovery of cell proliferation, indicating that CDK2 is a critical mediator of this response (Figure 6).
4. These findings were corroborated with both INX-315 and PF-4091 CDK2 inhibitors illustrating that this phenomenon is not drug selective and indicates that both drug actions are largely mediated by CDK2.

Selectivity of INX-315: Given that CDK1 is a well-known mediator of 4N arrest and pan-essential gene, we provide substantial new evidence, demonstrating that the cell cycle effects of CDK2 inhibitors, INX-315 PF-4091 are primarily mediated through CDK2 inhibition and not CDK1

1. *In vitro* kinase reactions reveal that the INX-315 is significantly more potent in inhibiting the catalytic activity of CDK2 as compared to CDK1 (Fig. 5g & 5H).
2. Distinct cellular responses are shown between INX-315 and a CDK1-selective further highlighting selectivity toward CDK2 for function (Fig. S6B).

3. Depletion of CDK1, CCNB1 and FOXM1 enhances the efficacy of pharmacological CDK2 inhibitors, suggesting the functional role of CDK1 pathway in augmenting the CDK2-inhibitor-mediated cell cycle arrest (Figure 7).

Exploring Biomarkers predicting response to CDK2 inhibition in breast cancer: The reviewer requested more details related to the biomarker analyses, which has been significantly expanded in response to their request.

1. The exceptional response to CDK2 inhibition occurs in tumor models that harbor high levels of P16INK4A and cyclin E1 (Figure 3).
2. Depletion of CDKN2A encoding P16INK4A is sufficient to overcome the G1 arrest in exceptionally sensitive models (Fig. 3D & 3E).
3. The concurrent expressions of *CDKN2A* and *CCNE1* occur in breast cancer tumor samples as demonstrated from clinical studies, including METABRIC, PEARL and PALOMA-2 (Fig. 4D & 4F).
4. The high expression of *CDKN2A* and *CCNE1* is associated with the therapeutic resistance as illustrated by association with the Basal-like ER+ breast cancer subtype and elevated expression with progression on CDK4/6 inhibitors (Fig. 4G)

Reviewer #1 - cell cycle regulation (Remarks to the Author):

This is an interesting analysis of the dependence of CDK2 and the effect of the CDK2 inhibitor INX-315 in human cancer cells. As previously known, few cancer cells display a strong dependence on CDK2. By using a combination of RNA interference and INX, the authors propose that co-expression of P16INK4A and cyclin E may be useful as biomarkers for response to CDK2 inhibitors. Accordingly, CDK4 and CDK2 ablation or inhibition cooperate in specific models. By studying cell lines with lower response to CDK2 inhibitors, additional phenotypes such as 4N arrest are observed when using higher doses of the inhibitor. Since these phenotypes are not observed upon CDK2 genetic ablation, a hypothesis is proposed that suggests additional effects of CDK2 (more obvious when CDK4 is also inhibited) in other phases of the cell cycle.

We thank the reviewer's positive feedback on our manuscript. We have carefully considered all the reviewer's comments and made revisions accordingly. A detailed response to each comment is provided below.

The study is very well executed and there are many interesting observations and conclusions in the first part of the manuscript, although novelty perhaps is more reduced. Data in the second part (Mechanistic impact of CDK2 inhibition beyond G1/s) are less solid, leading to mechanistic conclusions that are mostly speculative. I would suggest to moderating most of the conclusions in this section unless the authors want to investigate more in detail these phenotypes.

We apologize for any lack of clarity in the conclusions or for overstating the interpretations in the initial version of our manuscript. As suggested by the reviewer we have moderated the conclusions and addressed their specific comments with the inclusion of new data as detailed below.

Specific comments and questions:

The authors suggest senescence in MB157 cells citing ref. 38. Whereas Fig. S2A look senescent, it may be important to verify this, especially when discussing increasing DNA content, a phenotype also observed as a consequence of the DNA damage response (see below).

As suggested by the reviewer we interrogated the senescent phenotype in MB157 cells following the treatment with INX-315. Our data illustrated increased senescence-associated beta-galactosidase (Fig. S2C). We also determined whether the increase in DNA content following INX-315 treatment was associated with DNA damage by examining the γ H2AX foci formation (Fig. S2D). INX-315 did not induce γ H2AX, while the CHK inhibitor, CHIR124, which was used as a positive control, induced γ H2AX under the same conditions (Fig. S2D).

There is problem in the manuscript with the use of “4N” or “G2/M” arrest. These concepts are used interchangeably in the manuscript, leading to possible misinterpretations. For instance, the authors use these data to support that cells are senescent (G1 4N arrest; several sections of the text and Ref. 38) or to support the hypothesis that arrested cells are in G2/M (lines 424-425; 434-435; Fig. 5E) supporting the effect of these inhibitors in other phases of the cell cycle (lines 435-438, 490-491, abstract, etc.). It is actually quite possible that all these cells are arrested in 4N G1, but the only way to discriminate between 4N G1 and G2/M is to use additional G2 or prometaphase markers together with DNA content.

The reviewer raises an important point regarding the 4N arrest. In response, we performed additional experiments to better define the cell cycle phases following INX-315 treatment. Based on flow cytometry and biochemical analysis we conclude that the 4N arrest induced by INX-315 in MB157 cells correspond to a pseudo-G1 like arrest (Fig. 2A, 2 B & 2C). Western blot indicates that RB is dephosphorylated and downstream cell cycle target proteins Cyclin A and Cyclin B1 are suppressed in the presence of INX-315, which are the indicative markers for G1-arrest (Fig. 2C). This phenomenon is also apparent with *CCNA2* deletion in MB157 cells (Fig. 1H & 1I)

As a complementary approach, we employed live cell imaging using the PIP-FUCCI reporter system in MB157 cells to visualize the cell cycle phases in real time (PMID: 30421640). In this system the G1 phase cells express mVENUS, cells at S-phase express mCHERRY and the G2/M phase express both the reporters (Yellow fluorescence). Single cell tracking indicated that although cells entered M phase, they cannot divide rather re-entered G1-like phase and underwent arrest (Fig. 2D & Fig. S2B).

This response contrasts with tumor models that are independent of the CDK2 gene (ie. MCF7, MiaPaCa-2, and 3226). Biochemical analysis reveals that the INX-315-mediated 4N arrest is associated with an increase in the inhibitory phosphorylation of CDK1 at the Y15 residue, continued phosphorylation of RB, and accumulation of cyclin B1 expression in MCF7, 3226 and MiaPaCa-2 cell lines (Fig. 5C).

In conclusion, the 4N block induced by INX-315 indicates either a G1 or G2/M arrest, depending on the cellular context. According to our data, the CDK2 addicted models (e.g. MB157) undergo a 4N G1 arrest, while the CDK2 independent models undergo a 4N G2/M arrest. Furthermore, based on biochemical data from Fig. 8G, we conclude that the combination of INX-315 with palbociclib induces a G1 arrest with 2N and 4N DNA content along with replication collapse. In

the revised manuscript we consistently used the term “4N arrest”, unless the nature of the arrest was defined by additional experiments.

The authors claim that “INX-315 suppresses proliferation through a mechanism that involves CDK2-mediated cyclin sequestration”. I would say that data in the manuscript do not support this conclusion, or that this possibility has not really been addressed in the manuscript. For instance, lines 371-374 indicate that the effect of high doses of INX is reversed by depletion of CDK2. However, this is only true for MiaPaCa (which arrested in G1), but not for MCF7 (with arrest in G2/M)(Fig. 4B,C).

We acknowledge that our previous conclusion was overstated with limited supporting data. In the revised manuscript we have moderated our conclusions based on additional data that defines the mechanism of INX-315: The key conclusions are as follows

1. There is a distinct cellular response between genetic depletion of CDK2 and pharmacological inhibition with INX-315.
2. INX-315 induces cell cycle arrest with 4N DNA content and enhances the G2/M markers such as CDK1 phosphorylation and cyclin B1 expression (Fig. 5C).
3. Unbiased analyses revealed the CDK2 is a key determinant of sensitivity to INX-315 in diverse cell models (Fig. 6A & B).
4. The 4N block following INX-315 and PF-4091 treatment is CDK2 dependent, which is shown in different cell lines (Figure 6).
5. The cellular response to INX-315 is distinct from the pharmacological inhibition of CDK1 (Fig. S6B, Fig. 6J & Fig. S6E).

Based on these findings we re-wrote the text to highlight the importance of CDK2 as the predominant binding target for the pharmacological inhibitors, INX-315 and PF-4091. The distinct response between CDK2 deletion and inhibitors suggest that the CDK2 complex bound to an inhibitor could act via multiple mechanisms to inhibit cell cycle progression and defining such mechanisms require further investigation.

In agreement, whereas CDK1 complexes did not change in MiaPaCa, it seems that CDK1 is increasingly bound to Cyclin B1 in MCF7 (in the only WB-IP shown; CDK1 levels decrease whereas Cyclin B1 increase; replicates of this experiment should be carefully quantified and presented).

As suggested by the reviewer, we repeated the CDK2 and CDK1 immunoprecipitation in both MiaPaCa-2 and MCF7 cells three independent times and we quantified the fraction of cyclins bound to CDKs normalized to the input (Fig. 5F). We conclude that INX-315 increases the formation of CDK2/Cyclin E and CDK2/Cyclin A complexes in both MCF7 and MiaPaCa-2 cells (Fig 5E & 5F). Although INX-315 increases the expression of cyclin B1, the fraction of cyclin B1 in complex with CDK2 and CDK1 remains low (Fig. 5F & Fig. S5D). In the revised manuscript we have discussed this observation.

Together with other reasons discussed in the manuscript, it is very difficult to discriminate to what extent G2/M-4N G1 arrest depends on cyclin sequestration or partial CDK1 inhibition. In the absence of CDK2, very partial inhibition of CDK1 may have strong effects. Most of the assays in the manuscript (e.g. Fig. 4C,D,E,F, or the subsequent analysis of G2/M genes) are not useful to discriminate between the hypothesis of cyclin sequestration, favored by the authors, versus partial effects in CDK1. The fact that INX-315 induces G1 arrest at low doses and 4n arrest at higher doses “in the same cell line” (e.g. Fig. 2A), actually favors the option of inhibition of additional kinases at higher doses. In the same cell line, CDK2 inhibition and cyclin sequestration should still arrest first in G1, whereas 4n arrest could easily be explained by CDK1 inhibition.

We agree that differentiating a CDK2-mediated G2 arrest from effects on CDK1 is critical and challenging. Multiple data in the revised manuscript indicate that CDK2 is the important gene target for responses to INX-315 and PF-4091.

In response to the reviewer’s comment, we rephrased our conclusions regarding the CDK/Cyclin complexes in the revised manuscript based on co-immunoprecipitation assays (Fig. 5E, 5F & Fig. S5D). Additionally, we evaluated the impact of INX-315 on CDK1 kinase activity. In the revised manuscript we included an *in-vitro* kinase reaction, which demonstrates that the efficacy of INX-315 is significantly more potent against CDK2 kinase than CDK1 (Fig. 5G & 5H).

Moreover, our data demonstrates that in the absence of CDK2, the cellular outcomes of INX-315, induction of 4N block, tyrosine phosphorylation of CDK1 and inhibition of cell proliferation were diminished across three different cell lines (MCF7, MiaPaCa-2 and 3226) (Figure 6, Fig. S6C, S6D). These observations imply that off-target effect of INX-315, for instance, CDK1 inhibition, cannot fully account for its cytostatic effect in 4N as the drug’s activity is largely dependent on CDK2.

Comparing the effect of INX-315 vs. CDK1 inhibition further reinforces that if there are “off target” effects they are likely not leading to the G2/M blockade through CDK1 (Fig. S6B).

Moreover, the combination with Aurora/FoxM1 etc. inhibition does not support the hypothesis of G2 functions as these treatments also cooperate with palbociclib in multiple tumor scenarios. To analyze this in most detail, Fig. 4F should be performed with CDK1-Cyclin A2 and CDK1-Cyclin B1 in parallel to CDK2.

To address the reviewer’s comment, we compared the effect of palbociclib and INX-315 following FOXM1 depletion in MiaPaCa-2 and HCC1806 cell lines. Our data indicates that FOXM1 depletion selectively enhances the efficacy of CDK2 inhibition with minimal impact on CDK4/6 inhibition in both the cell lines (Fig. S10D, Fig. 7B & 7C).

The BLI assay (formerly Fig 4F) cannot be utilized to determine specificity as the assays are carried out in the absence of ATP or other proteins. Rather that assay is used to specifically monitor the effect of the drug on complex stability. We have rephrased the discussion of the assay which illustrates that as a Type 1 kinase inhibitor INX-315 can stabilize the CDK/Cyclin complex which is consistent with our findings with CDK2 complexes by co-IP (Fig. S5E & Fig. 5F).

To determine selectivity, we employed *in vitro* kinase reaction in the revised manuscript further corroborates the high selectivity of INX-315 for CDK2 over CDK1 (Fig. 5G & 5H). We also included data from the development of INX-315 which evaluated the cellular IC50 vs. CDK1 and CDK2 assembled kinase complexes (Fig S5F).

CDK1 depletion could also be tested although it is understandable that these assays depleting CDK1 may be difficult due to the essentiality of this gene (that may also explain why does not show up as a hit in Fig. 4B. It would be interesting to know the position of CDK1 and other CDKs or CDK-like kinases in that plot). Can the authors provide the results of the CRISPR screening in Fig. 4B (ranking of genes/sgRNAs) as Supplementary Information?

This is a very good suggestion by the reviewer to evaluate the impact of CDK1 deletion on the cellular response to INX-315. In response to reviewer's comment, we did a transient CDK1 knockdown using RNAi in two different cell lines (MiaPaCa-2 and HCC1806) to demonstrate a cooperative effect in combination with INX-315 (Fig. 7F & 7G).

As suggested by the reviewer we have modified our CRISPR screen results. We included the genes that are significant in both the cell lines (Fig. 6A, & 7A). The entire data set is uploaded as Supplementary Information.

In general, it is not easy to understand why the authors think that higher doses of INX have different phenotypes by uniquely affecting the very same kinase. The IC₅₀ for CDK2-CycE/A is in the range of 0.6-2.5 nM and the IC₅₀ for CDK1-CycB is about 30 nM (10-fold; about the same difference than cells sensitive to 30 nM vs. 500 nM). In addition, I did not find data for the IC₅₀ on CDK1-CycA but it will very likely be more similar to CDK2-CycA. As discussed above, demonstrating that these effects at high doses do not depend on CDK1 inhibition would require direct evidence rather than indirect correlations.

We thank the reviewer for raising the question regarding the IC₅₀ of INX-315 across different cyclin/CDK complexes. In the revised manuscript we included a table summarizing the IC₅₀ values based on the previously published study (PMID: 38047585) (Fig. S2A & S5F). We also evaluated the IC₅₀ of INX-315 against CDK1/Cyclin A (540 nM), which is at least 8-fold higher than that against CDK2/Cyclin A (Fig. S5F).

These data demonstrate that INX-315 exhibits higher affinity for the CDK2/Cyclin E1 complex over CDK2/Cyclin A (Fig. S2A). In cell culture studies, INX-315 was exceptionally effective at the concentration range 30-50 nM in *CCNE1/CDK2* addicted models like MB157 and KURAMOCHI (Fig. 2E & Fig. S2F). In these models, *CCNA2* depletion yields 4N-G1 arrest, a phenomenon similar to the higher dose of INX-315 (Fig. 1H & 2B). A CDK1-selective inhibitor, RO-3306, exhibited distinct cellular response as compared to INX-315 (Fig. S6B). Furthermore, the impact of RO-3306 on cell proliferation and the induction of 4N block cannot be reversed following CDK2 deletion (Fig. 6J & Fig. S6E). These findings provide further evidence that the cellular effects of INX-315 are dependent on CDK2 inhibition as opposed to CDK1.

Lines 344-346. It may be informative to indicate the percentage of tumors with these two markers.

As suggested by the reviewer, we have indicated the percentage of tumors that harbor high Cyclin E1 and P16INK4A in the revised manuscript.

Why using cyclin A1 (meiotic cyclin) for biochemical studies whereas cyclin A2 is targeted in the genetic assays?

We regret this error. The BLI assay was performed using cyclin A (encoded by CCNA2) for the interaction with CDK2.

Is it a bit suppressing that cyclin A is not bound to CDK1 in Fig. 4E. Does this antibody recognize CCNA1, CCNA2 or both?

The anti-cyclin A antibody is highly specific to Cyclin A, which we have validated by depleting CCNA2 using siRNA (Fig. 1I). Based on our data, there is an interaction between cyclin A/CDK1, however, when we compare with the total cyclin A expression in the input, the fraction bound to CDK1 is low in both MCF7 and MiaPaCa-2 cell lines (Fig. S5D & Fig. 5E).

Very difficult to see labels in some critical plots (Fig. 2E,G, 3A, 4B, 5A, 5D, etc) References 21 and 23 are duplicated. Some references are incomplete (8, 18, 31, 41).

We thank the reviewer for pointing out these issues. We have addressed them in our revised manuscript and ensured that all references are accurate and appropriate.

Reviewer #2 - CDKs, genetic screens (Remarks to the Author):

This study focuses on the outcomes of CDK2 inhibition in human cancer cells. The subject of the study is very timely, as CDK2 inhibitors are now used in several clinical trials. The authors identified “CDK2-addicted cell lines” (exceptional responders) in which CDK2 inhibition arrested cell growth and cell lines relatively insensitive to CDK2 inhibition (called by authors “cluster 2” cells). The main take-home messages of the study are: (1) “CDK2-addicted cell lines” have high P16INK4A expression. This indicates that P16INK4A expression can be used as a predictor of a strong response to CDK2 inhibitors; (2) Using CDK2 inhibitor INX-315, they performed a screen for knockouts that make cells resistant to CDK2 inhibition and identified CDK2 as a top hit; (3) In a screen for knockouts that sensitize cells to CDK2 inhibition they identified G2-specific genes, such as FOXM1 or cyclin B. (4) They demonstrate a strong synergistic effects of CDK2 and CDK4 inhibition (Palbociclib) in cancer cell lines and patient-derived xenografts .

Observation #1 is very novel, interesting and of important clinical value. Also observation #3 is interesting and unexpected. Regarding #2, I found it very confusing that knockout of CDK2 (which should phenocopy CDK2 inhibition) actually makes cells resistant to CDK2 inhibition.

We agree with the reviewer that the genetic deletion of CDK2 would be expected to phenocopy CDK2 inhibition. However, our study identified through an unbiased genome-wide CRISPR screen that CDK2 loss drives resistance to INX-315. This finding was further validated across different cell lines using RNAi and CRISPR-mediated approaches. Additionally, we confirmed this resistance using another CDK2 inhibitor, PF-4091. A similar phenomenon where the genetic deletion does not phenocopy kinase inhibition has been reported in literature (PMID: 20953138; 32927473)

Overall, our data in the revised manuscript demonstrates that the efficacy of CDK2 inhibitors beyond G1 arrest largely depends on the presence of CDK2, highlighting that the cellular effects of these inhibitors rely on their ability to bind and inhibit their specific cellular target.

The authors show that INX-315 stabilizes cyclin-CDK2 complexes. Does it mean that CDK2 acts as an inhibitor of cyclins?.

Our data shows that INX-315 significantly enhances the formation of Cyclin E1/CDK2 and cyclin A/CDK2 complexes in two different cell lines, MiaPaCa-2 and MCF7 (Fig. 5F). Through additional experiments we conclude that CDK2 binds with available cyclin A and cyclin E1 as minimal protein was detected in the flow through following the pull down of CDK2 (Fig. 5E).

The reviewer raises an interesting point regarding Cyclin inhibition; however, there are alternative mechanisms, such as sequestration of various co-factors or interference with critical substrate accessibility might contribute to cell cycle arrest. Defining such mechanisms, however, is beyond the scope of this study. Therefore, we have moderated the conclusions to align more precisely with our data related to the CDK2-mediated mechanism.

The authors propose that CDK2 inhibitor may “trap” cyclin A in an inactive complex, whereas CDK2 knockout might allow formation of cyclin A-CDK1. They may be true, but needs to be demonstrated.

We recognize that the term “cyclin A trapping” is an overstatement and have rephrased the discussion. We emphasize our conclusions that the treatment with INX-315 inhibits cell proliferation and enhances the Cyclin A/CDK2 complex formation (Fig. 5E & 5F). This suggests a plausible mechanism that the CDK2/Cyclin A complex in the presence of INX-315 could negatively impact cell proliferation, which is distinct from CDK2 deletion. Furthermore, CDK2 is essential to elicit cellular responses to INX-315 and PF-4091 (Figure 6).

To address the reviewer’s subsequent comment, we observed that CDK2 deletion did not promote the assembly of cyclin A/CDK1 complex in our models (data not shown). One potential mechanism driving cell proliferation in the absence of CDK2 may involve the compensatory role of CDK4 and 6 kinase as evidenced by the enhanced response to Palbociclib in the absence of CDK2 (Fig. S10A)

Observation #4 is very solid but not unexpected - inhibition of CDK4/6 plus CDK2 is expected to block division of cancer cells.

We agree with the reviewer’s comment. Our study mainly highlights a therapeutic opportunity for aggressive tumors that do not express high levels of P16INK4A (ie. PDAC) by concurrently targeting CDK4/6 and CDK2.

Specific points:

1. Fig. 2A. INX-315 at higher concentrations arrested “CDK2-addicted” cell lines in G2/M. Could this be due to off-target effects, for instance on CDK1? This needs to be addressed.

To address the reviewer’s comment, in the revised manuscript we demonstrated that INX-315 is significantly more selective for CDK2 over CDK1 in inhibiting catalytic activity (Fig. 5G & 5H). Additional experiments in MB157 cells revealed that the accumulation of cells with 4N DNA content in MB157 cells correspond to a pseudo-G1 arrest as evidenced by the inhibition of RB phosphorylation (Fig. 2C). Conversely CDK1 depletion leads to an increase in RB phosphorylation

suggesting that the cellular outcomes following INX-315 in MB157 cells are not mediated through CDK1 inhibition (S1G).

To further demonstrate the intracellular target specificity, we deleted CDK2 using RNAi and CRISPR-CAS9 approaches, which mitigated the cellular effects of INX-315 on 4N arrest and molecular targets, confirming that CDK2 is the primary intracellular target. This has been demonstrated by a new Figure: 6

Additionally, a CDK1 selective inhibitor RO-3306, elicits distinct cellular responses as compared with INX-315 (Fig. S6B).

2. Similarly, “cluster 2” cell lines eventually arrest cell proliferation if sufficiently high concentration of CDK2 inhibitors are used – could this be an off target effect?

In response to the reviewer’s comment, we included additional data to define the cellular responses of cell lines which can proliferate in the absence of CDK2 to INX-315. Our biochemical analysis shows that INX-315 increases the inhibitory phosphorylation of CDK1 and accumulation of cyclin B1 (Fig. 5C). This phenomenon is abrogated by CDK2 depletion/deletion (Figure 6 & Fig. S6D). In total, our data indicates that CDK2 is the key mediator of these responses.

3. Along the same lines: Knockout of RB1 in “CDK2-addicted” cell lines makes them insensitive to low concentration of CDK2 inhibitor, but these cells still respond to higher concentrations (Fig. 2F). Could this be off-target effect?

To address the reviewer’s comment, we compared the molecular changes between the CDK2 addicted MB157-WT and the RB-del cell lines following the treatment with high concentrations of INX-315 (250 and 500 nM) (Fig. 5D). In the MB157-WT cells, INX-315 inhibited RB phosphorylation and repressed the cell cycle proteins, cyclin A, cyclin B1 and CDK1 (Fig. 5D). However, in the MB157-RB-del cells, INX-315 results in enhanced phospho-CDK1 and cyclin B1 (Fig. 5D).

4. Fig. 1D. Inhibition of CDK2 inhibited expression of cyclins A and B1. This is likely consequence of cell cycle arrest – this should be clearly stated.

We agree with the reviewer that Fig. 1E in the revised version (formerly Fig. 1D), CDK2 depletion leads to the downregulation of cyclin A and cyclin B1 in the CDK2-addicted models, MB157 and KURAMOCHI, which are regulated by E2F indicating cell cycle arrest. In the revised manuscript we have stated this clearly.

5. In the part about high P16INK4A levels conferring sensitivity to CDK2 inhibition. The authors overexpressed P16INK4A and show that this makes cells sensitive to CDK2 inhibition. But P16INK4A overexpression already had a strong effect on cell growth (Fig S3C).

We apologize for the lack of clarity while discussing this data. Ectopic overexpression of P16INK4A potently suppressed the proliferation of MCF7-WT cells (Fig. S4D). However, we demonstrate that in MCF7 cells, overexpressing cyclin E1 (MCF7-E1), P16INK4A only partially inhibited cell proliferation, which was further enhanced in the presence of INX-315 (Fig. S4D).

It would be beneficial to evaluate how the genetic depletion of CDKN2A will affect response to CDK2 inhibition. I would suggest employing a CRISPR-Cas9 mediated knockout of CDKN2A in 2-3 models belonging to 'cluster 3' to evaluate this hypothesis.

In our current study, we mainly employed MB157 and KURAMOCHI cell lines from cluster 3. As suggested by the reviewer we evaluated the impact of *CDKN2A* depletion on the cellular response to CDK2 inhibition in both KURAMOCHI and MB157 cells. Following the RNAi-mediated knockdown of *CDKN2A*, the efficacy of INX-315 is mitigated in KURAMOCHI cells (Fig. 3D & E). However, CDKN2A deletion did not impact the cellular response to INX-315 in MB157 cells which is likely due to the higher expression of other INK4 family genes (*CDKN2B*, *CDKN2C*, *CDKN2D*) in MB157 cells (not shown). Therefore, we hypothesized that CDK4 overexpression could bypass the inhibitory effects of these composite proteins and drive resistance to INX-315. Ectopic overexpression of CDK4 mediated resistance to INX-315 in both KURAMOCHI and MB157 cells (Fig. 3F & 3G).

6. In Fig.5G authors have shown how Palbociclib and INX-315 treatment changes the protein level of cell cycle components. It is surprising that treatment with Palbociclib in a CDK4-dependent model - MCF7 - does not lead to downregulation of RB1 phosphorylation in the presented data.

We agree with the reviewer, MCF7 cells exhibit sensitivity to CDK4/6 inhibitors. For the combination treatment with INX-315, we deliberately used low dose of palbociclib (50 nM) to highlight that the sub-optimal impact of CDK4/6 inhibition on RB activation and suppression of cell cycle proteins could be enhanced by co-targeting CDK2 (Fig. 8G). The synergistic interaction between Palbociclib and INX-315 in MCF7 cells was illustrated based on BrdU incorporation assay (Fig. 8F).

7. Most of experiments were done with a CDK2 inhibitor INX-315. It would be good to confirm at least some key results with other CDK2 inhibitors.

This is a very good suggestion by the reviewer. In the revised manuscript we incorporated another CDK2 inhibitor, PF-4091 to further validate our key findings. We observed that the impact of PF-4091 is comparable to that of INX-315 (Fig. S5A, Fig. 6G, 6H, 7C, 7E). Therefore, the cellular responses we observed are a direct consequence of CDK2 inhibition rather than being specific to a particular inhibitor.

Reviewer #3 - Breast cancer, CDKs (Remarks to the Author):

Kumarasamy et al describe vulnerabilities and biomarkers of CDK2 inhibition using cancer cell line models. An exceptional sensitivity to CDK2 inhibition (INX-315) is conferred through a combination of RB-dependent transcriptional responses and the expression of P16INK4A/INK4A. In other cell lines, the CDK2 inhibitor INX-315 suppresses proliferation through a mechanism that involves CDK2-mediated cyclin sequestration. Cooperation of CDK2 plus CDK4/6 inhibition is also studied.

The study is well performed and rises some questions that would be worth addressing.

Figure 1A is a graphical representation of a DEPMAP analysis showing vulnerabilities to cell cycle kinases (CDK2/4/6) and cyclins (D1/2/3 & E1). Would it be interesting to show a lane including cyclin A, since it also complexes with CDK2? Is there an overlap of cell lines with CDK2 / cyclin A lethality or is it a different cluster to the one of CDK2 / cyclin E1?

As suggested by the reviewer we included the CHRONOS score to define the vulnerability to loss of *CCNA2* (Fig. 1A). As illustrated in our data, every cell line in each cluster is dependent on *CCNA2*. This is further validated using RNAi-mediated, where deletion of *CCNA2* inhibited cell proliferation in multiple models that are intrinsically independent of the *CDK2* gene (Fig. 1F). Mechanistically *CCNA2*-mediated growth arrest occurs due to the cell cycle arrest with 4N DNA content (Fig. 1G).

Also in this Figure 1A, it would be interesting to indicate the type of cancer cell line right below the cluster marks: breast (ER+, TNBC, HER2+), ovarian, gastric, etc... to capture any disease-enrichment within the clusters; e.g. is cluster 2 enriched in ER+ BC and cluster 3 enriched in ovarian cancer and TNBC cell lines?

This is a very good suggestion; however, due to the number of cell lines it is very hard to see in the heatmap. In the revised manuscript, we have included a new heat map (Fig 1B) that depicts the enrichment of different tumor types in each cluster. Furthermore, as suggested we divide the breast cancer cell lines by "subtype" which is summarized in Figure S1A.

The volcano plots barely readable.

We regret the annotations in our volcano plots. In the revised manuscript, we have fixed this issue.

Figure 3A shows a cluster comparison btw cluster 3 vs cluster 2 where *CDKN2A* and *CCNE1* appear to be the enriched factors btw CDK2/cyclin E1- vs CDK4/cyclin D1-dependent clusters. Does the comparison btw cluster 3 vs cluster 6 (CDK6/cyclin D1-dependent), also render *CDKN2A* as differential factor?

In response to the reviewer's comment, we analyzed the differential gene expression between the Cluster 3 (*CCNE1*/CDK2-dependent) and cluster 6. Based on our volcano plot, the expressions of *CDKN2A* and *CCNE1* are significantly higher in cluster 3 as compared to that in cluster 6 (Fig. 3A).

Figures 3C/D show how overexpression of *P16INK4A* renders HCC1806 cells to be dependent on CDK2 for pRb phosphorylation. Does *P16INK4A* overexpression also sensitize HCC1806 cells to CDK2i?

In the revised manuscript, Fig. 3C demonstrates that ectopic expression of *P16INK4A* in HCC1806 cell line renders this model to be more sensitive to INX-315 as indicated by enhanced inhibition on RB phosphorylation and downregulation of cyclin A.

Figure 3G shows two examples of TNBC tumors with high P16INK4A and cyclin E1. Figure 3H shows an analysis of coexpression of biomarkers. How many tumors out of 139 TNBCs were high P16INK4A and cyclin E1 protein expression that are RB positive.

In response to the reviewer's comment, we have included a scatter plot in the revised manuscript that illustrates the correlation between the expression of P16INK4A and cyclin E1 based on the multi-spectral staining on the TNBC TMA, comprising 197 tissue samples. We also characterized the tissues as RB proficient and RB deficient. Among these, 21 tumors were found to be RB proficient and displayed high levels of P16INK4A and Cyclin E1 (Fig. 4A & 4B).

Any chance to show data from ER+ BC tumours? Several clinical trials have reported expression or protein data: e.g. PALOMA2/3 (PMID: 35974168) and PEARL (PMID: 33385521) have run the the HTG Edge-Seq Oncology Biomarker Panel; the Preoperative trials (POP, PMID: 29893769 and Abema-POP; PMID: 36071033) from Institute Gustave Roussy have run IHC of these biomarkers in metastatic/primary ER+BC.

As suggested by the reviewer we included data from ER+ breast cancer patients to investigate the co-expression of P16INK4A and cyclin E1. Analysis of METABRIC dataset reveals that only a small subset of ER+ samples express high levels of both P16INK4A and cyclin E1 as compared with TNBC (Fig. 4D). This is not unexpected and supports the better efficacy of CDK4/6 inhibitors in ER+ breast cancer vs. TNBC. Importantly, basal-like ER+ tumors are those that express high-levels of CDKN2A and CCNE1 (Fig. 4E). This would agree with findings that the basal-like tumors have limited/no benefit from treatment with CDK4/6 inhibitors clinically (PMID: 33769862).

Analysis from PALOMA-2 and PEARL trial data reveals a positive correlation between CCNE1 and CDKN2A (Fig 4F). However, there are very few basal like tumors and not a distinct CCNE1/CDKN2A high population.

We analyzed our own ER+ breast cancer patient cohort (NCT04526587), comparing the expression of CCNE1 and CDKN2A before and after progression with CDK4/6 inhibitor-based treatment (PMID: 37704753). Our results indicate a significant increase in CCNE1 expression post-treatment while the general CDKN2A is unchanged (Fig. 4G). However, there are two patients with elevated CDKN2A and high CCNE1 expressions which exhibited very short progression-free survival (Fig. 4G). These data in total underscore that while high co-expression of CDKN2A and CCNE1 can occur in ER+ breast cancers, it is relatively rare but generally associated with CDK4/6-inhibitor resistance.

Figure 4A shows sensitivity of the ER+ BC cell line MCF7 to high-dose CDK2i. Is the high-dose CDK2i tolerable for in vivo use in mice? Please provide dose response of MCF7 xenografts.

As recommended by the reviewer we conducted additional *in vivo* experiments using MCF7 xenograft models. Mice were treated with a higher dose of INX-315 (200 mg/kg), which was well tolerated as no change in the body weight was observed (Fig. 9B). INX-315 significantly delayed tumor growth and demonstrated potent disease control compared to the vehicle-treated group (Fig. 9A & 9C).

Figure 4A. Please for consistency, provide experiment for HCC1806

As suggested by the reviewer, we evaluated the effect of CDK2 inhibitors, INX-315 and PF-4091 on the proliferation of HCC1806 cells following the treatment with different concentrations of INX-315. (Fig. 5A & Fig. S5A)

Figure 4B. What are the other hits positively selected following INX-315 treatment? Any specific cyclin? Is RB1 found? Please provide list as supplementary table(s).

In response to the reviewer's comment, we have incorporated additional genes that were positively enriched following the selection with INX-315 in MiaPaCa-2 and HCC1806 cell lines. We found loss of AMBRA1 and DYRK1A, which are negative regulators of cyclin D1, are associated with resistance (Fig. 6A). RB1 was not identified as driver for resistance to INX-315 in these models, which is consistent with the sustained RB phosphorylation in the arrested cells (Fig. 5C). We have included the entire list in the supplementary tables.

Figure 4E. Please provide evidence that CDK2 kinase activity is reduced upon treatment with INX-315.

As suggested by the reviewer, we included additional data where we evaluated the catalytic activity of CDK2 in the absence and presence of different concentrations of INX-315 using *in vitro* kinase reactions. INX-315 is highly potent in inhibiting the catalytic activity of CDK2 as determined by the phosphorylation status of the exogenous substrate, which is a C-terminal peptide from RB (Fig. 5G).

Figure 5E. Please reason why are there different concentrations of INX-315 and palbo used.

We choose the concentrations of Palbociclib and INX-315 based on the synergy experiments, where we have used multiple doses of both the drugs (Fig. 8F). We chose the dose where the single agent possesses modest cellular impact.

Please show consistent cell cycle color coding across the manuscript

We regret the inconsistencies. In the revised manuscript we have fixed this issue.

Figure 6D. please cite in the text; the figure is not mentioned.

We incorporated the citation for all the figures in the main text.

In the discussion (approx. line 480), it would be interesting to reason that P16INK4A has been identified as mechanism of resistance to CDK4/6i in ER+ BC, so P16INK4A can help identify patients with ER+ tumors that might be sensitive to target with CDK2i (PMID: 36071033).

There have been discordant interpretations on the expression of P16INK4A and resistance to palbociclib, which we now discuss. One school of thought was that the complexes enabled the kinase to remain active in the presence of the CDK4/6 inhibitor (PMID: 36071033), while others suggested that this defined a CDK4/6 inhibited state where the cell cycle was driven by other kinases (PMID: 34544752). The data here supports that in those tumors that express high levels of P16INK4A there is a dependence on CDK2 presumably because CDK4/6 is inhibited.

The discussion states (line 501): “The CRISPR screens identified genes that regulate CDK1 (e.g. CCNB1) and coordinate G2/M progression (FOXM1) that potentially cooperate with CDK2 inhibition. Presumably, drug screens identified aurora kinase inhibitors that perturb G2/M progression to cooperate with the CDK2 inhibitor, INX-315.” Were any chemotherapeutic agents present in the drug screens and provided rationale for combination approaches that are useful for the clinical development of CDK2i?

Chemotherapeutic agents were not identified to cooperate with INX-315 based on our drug screens (Fig. S8 & S9). The AURK and CDK inhibitors that impact the G2/M progression were identified from our drug screens.

We thank the reviewers for the positive feedback on our revised manuscript. Below is our response. The reviewer's text is copied verbatim, and our response is provided in blue font.

Reviewer #1 (Remarks to the Author):

The authors have made a significant effort to address all reviewers' comments and the manuscript is much more solid now. It describes very interesting data using solid and elegant assays, provides nice discussion and data may have clinical potential. I still believe that the three evidences provided to indicate the selectivity of INX-315 (1) more potent on CDK2 than CDK1; 2) distinct responses when compared to CDK1; c) cooperation with CDK1, CCNB1 or FOXM1 ablation) do not exclude the possibility that INX315, at high doses, induces 4N arrest by ""partially"" inhibiting CDK1 as well as CDK2 kinase activity in a direct manner. Yet, I understand the difficulties of discriminating between direct and indirect partial inhibition of CDK1.

We thank the reviewer for the comments. We agree with the reviewer that at high concentrations (500 nM-1000 nM) INX-315 partially inhibits CDK1.

Reviewer #2 (Remarks to the Author):

My concerns have been satisfactorily addressed.

We are glad that our responses satisfied the reviewer's concerns.

Reviewer #3 (Remarks to the Author):

The authors have addressed the reviewer questions.

We thank the reviewer for accepting our responses.